# Forests, Water, and Land Use Change across the Central American Isthmus: Mapping the Evidence Base for Terrestrial Holocene Palaeoenvironmental Proxies

**William J. Harvey** [1,2,3,*] **, Gillian Petrokofsky** [1,3] **, Nathan Stansell** [4] **, Sandra Nogué** [5] **, Leo Petrokofsky** [3] **and Katherine J. Willis** [1]

1. Long-Term Ecology Laboratory, Department of Zoology, University of Oxford, Oxford OX1 3SZ, UK; gillian.petrokofsky@oxsrev.org (G.P.); kathy.willis@zoo.ox.ac.uk (K.J.W.)
2. School of Archaeology, University of Oxford, Oxford OX1 2PG, UK
3. Oxford Systematic Reviews LLP, Oxford OX2 7DL, UK; leo.petrokofsky@oxsrev.org
4. Department of Geology and Environmental Geosciences, Northern Illinois University, De Kalb, IL 60115, USA; nstansell@niu.edu
5. School of Geography and Environmental Sciences, University of Southampton, Highfield, Southampton SO17 1BJ, UK; s.nogue-bosch@soton.ac.uk
* Correspondence: william.harvey@arch.ox.ac.uk

**Abstract:** An ever-increasing demand for agriculture while conserving biodiversity, maintaining livelihoods, and providing critical ecosystem services is one of the largest challenges for tropical land management across the Central American Isthmus today. Climatic and anthropogenic drivers threaten to cause changes in the forest cover and composition for this region, and therefore, understanding the dynamics of these systems and their variability across space and through time is important for discerning current and future responses. Such information is of value especially for risk mitigation, planning, and conservation purposes. The understanding of the forests, water, and land use for this region through time is currently limited, yet it is essential for understanding current patterns of change, particularly with reference to: (i) forest fragmentation; (ii) water availability; and (iii) land management. Through the examination of biotic (e.g., pollen, diatoms, and *Sporormiella*) and abiotic (e.g., $\delta^{18}O$, $CaCO_3$, and magnetic susceptibility) proxies, extracted from environmental archives, evidence for longer-term environmental changes can be inferred and linked to drivers of change including climate, burning, and human activities. Proxy environmental data from terrestrial depositional archives across the Central American Isthmus were identified and mapped following best practice for systematic evidence synthesis. Results from the evidence base were summarised to show the spatial and temporal extent of the published datasets. A total of 12,474 articles were identified by a comprehensive search in three major bibliographic databases. From these, 425 articles were assessed for relevance at full-text, and 149 fully met inclusion criteria for the review. These articles yielded 648 proxy records in 167 study sites that were mapped on an interactive map with filters to allow full exploration of the evidence base. Just under half of the studies were published in the last decade. Most studies extracted their data from lake sediments, with a focus on moist tropical forests in lowland sites in Guatemala, Belize, and Mexico. The largest data gaps in the evidence base are Honduras, Nicaragua, Panama, and El Salvador. There are also significant evidence gaps for dry tropical forests, coniferous forests, mangroves, and grasslands. Most of the studies assessed had methodological or presentational limitations that make future meta-analysis difficult and significantly affect the ability to draw conclusions that are helpful for future decision-making. A degree of standardisation, transparency, and repeatability in reporting would be beneficial to harness the findings of the existing evidence base and to shape future research in this geographical area. The systematic map of the evidence base highlights six key review topic areas that could be targeted, if the raw data could be obtained, including: (i) dating uncertainty and standardising reporting; (ii) land use change across space and time; (iii) dispersal pathways of agriculture; (iv) the role and impacts of fire and burning; (v) changes in hydro-climate, water availability, and the risk of tropical storms; and (vi) forest resilience and recovery.

**Keywords:** palaeo; environment; forest; Central America; systematic map

## 1. Introduction

The largest challenge for tropical land management across the Central American Isthmus is to meet the ever-growing demand for agriculture while conserving biodiversity, maintaining livelihoods, and providing critical ecosystem services [1]. Climatic and anthropogenic drivers currently provide the largest threats to changes in the vegetation cover and composition of this region. Understanding the dynamics of these systems, their variability across space and through time, and the impacts of climate and anthropogenic drivers on forests is therefore important for discerning current and future responses. For example, agricultural intensification has been accompanied by substantial reductions in tree cover, habitat diversity, and forest connectivity in recent years [2], with around 80% of the region's vegetation having been converted into agriculture [1]. Such information is of value especially for risk mitigation, planning, and conservation purposes [3].

In addition to human-driven land use change, identifying the potential changes in available water resources in response to climate change is another top priority of the IPCC and policy makers throughout the Americas [4]. At present, future drought is difficult to predict, as current projections for tropical circulation changes under a range of warming scenarios remain highly uncertain [5]. However, the impacts of droughts are a serious current problem. For example, a deficit in precipitation at the beginning of the primera cropping cycle (agricultural season) in 2015 caused significant losses in food production, rendering an estimated 2.2 million people at risk of moderate or severe food insecurity [6]. The habitat loss driven by human activity and climate change is leading to increased fragmentation of the remaining forests and consequent loss of biodiversity. For example, according to the IUCN Red List, over 300 of the region's endemic species of flora and fauna are currently threatened with extinction, of which 107 are critically endangered [7]. Research examining the drivers, and responses, of different vegetation types to climatic and anthropogenic influences across the Central American Isthmus predominantly focuses on anthropogenic impacts spanning the last 50 years [3]. Yet, many of the processes associated with forest change occur on timescales much longer than this. It is also unclear how much of the present landscape was already impacted by these drivers before 50 years ago and what, therefore, is its true "natural" baseline.

An understanding of the forests, water, and land use for this region through time is currently limited, yet it is essential for understanding current patterns of change, particularly with reference to: (i) forest fragmentation; (ii) water availability; and (iii) land management. Through the examination of biotic (e.g., pollen, diatoms, and *Sporormiella*) and abiotic (e.g., $\delta^{18}O$, $CaCO_3$, and magnetic susceptibility) proxies, extracted from environmental archives, evidence for longer-term environmental changes can be inferred and linked to drivers of change including climate, burning, and human activities.

### 1.1. Forests of the Central American Isthmus

Forests across the Central American Isthmus can be broadly described under six terrestrial biome types: (i) moist tropical forests; (ii) dry tropical forests; (iii) coniferous forests; (iv) grasslands, savannas and shrublands; (v) xeric shrublands; and (vi) mangroves (Figure 1).

Moist tropical forests are dominated by semi-evergreen and evergreen deciduous tree genera, such as *Alnus*, *Cassipourea*, and *Hevea* [9], and contain the highest tree species diversity of all the terrestrial biomes [3]. This biome can be further subdivided by ecoregions according to their altitudinal distribution, grouping them into lowland (<1000 m) and upland (>1000 m) areas [8]. The lowland moist broadleaf forests are distributed in continuous strips (e.g., the Petén-Veracruz moist forests and Central American Atlantic moist forests), whereas the montane moist tropical forests are naturally fragmented by

elevation (e.g., the Central American montane forests). The main drivers of land use change in this biome are agricultural and infrastructure expansion [10,11].

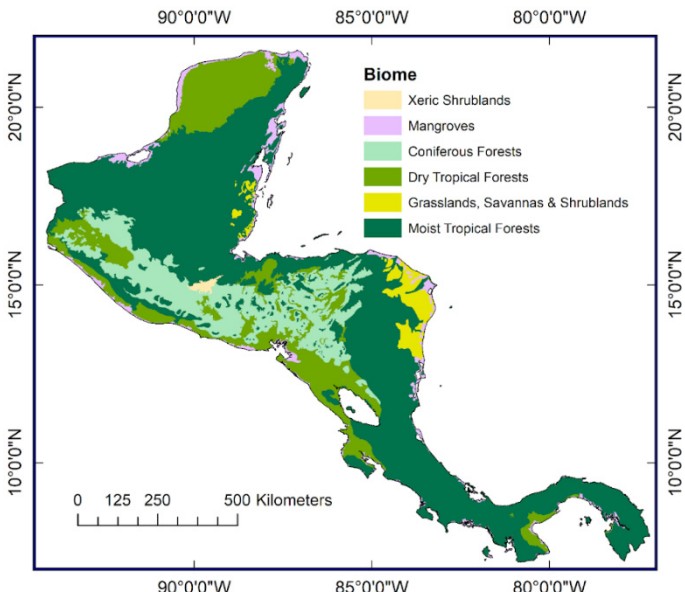

**Figure 1.** Biomes of the Central American Isthmus. Based on Dinerstein et al. [8].

Dry tropical forests were originally thought to have covered the north of the Yucatan Peninsula and extended in a continuous strip from the Pacific Coast of southern Chiapas, to north-western Costa Rica. These forests now only cover 1% of this area. This drastic reduction in size is thought to have been due to increasing human settlement and extensive agrarian practices [12]. This region is densely populated because it is ideal for shifting agriculture and cattle ranching [13]. These forests are usually composed of two strata: a higher story of deciduous trees, such as *Brosimum*, *Bursera* and *Cordia* [14]; and a lower story of evergreen species, such as *Diospyros*, *Mimosa* and Bignoniaceae [15].

Coniferous forests are mainly distributed across the uplands of the Sierra de Madre de Chiapas, México/Guatemala through the Sierra del Merendón, Guatemala/Honduras and south into northern Nicaragua. The most outstanding characteristic of this biome is the diversity of pine (>100 *Pinus* spp.) and oak (>150 *Quercus* spp.) species [16,17], which are well adapted to variable climatic conditions and natural fires [11]. These pine–oak forest formations often form intricate mosaics and complex successional interactions extending up into broadleaf cloud forests at higher altitudes [17,18]. This biome is currently threatened by agricultural expansion, logging, firewood extraction, forest fires, and pests [19].

Grasslands, savannas and shrublands can be found on the east coasts of Honduras and Nicaragua and in Belize. They predominantly comprise one species of pine, *Pinus caribaea*, and expansive grass fields. This ecosystem has poor soils and is subject to frequent burning [20].

Xeric shrublands are restricted to one ecoregion on the Guatemala–Honduras border. The Motagua Valley comprises two main vegetation types: (i) thorn scrubland, comprised of cacti, such as *Opuntia*; and (ii) dry forests comprised mainly of shrubby Leguminosae; however, there is a high diversity of tree communities in riparian habitats [11]. Most of this region has been converted into irrigated agricultural fields [21].

Mangroves grow along the coastal regions of Central America. Tree species are red mangrove (*Rhizophora mangle*), yellow mangrove (*Rhizophora harrisonii*), white mangrove (*Laguncularia racemosa*), and black mangrove (*Avicennia germinans*). Residential and commercial anthropogenic expansion into these areas is threatening biodiversity and soil stability [22].

### *1.2. Water and Climate*

Today, precipitation across the Central American Isthmus is principally driven by topography [23], as well as the complex interactions of factors influencing the Inter Tropical Convergence Zone (ITCZ) [24]. These include the North Atlantic Oscillation (NAO) and the El Niño Southern Oscillation (ENSO) [25]. Modern rainfall has a bimodal seasonal regime with high levels of precipitation between May and November and drier conditions between December and April [26,27]. Modest shifts in either eastern tropical Pacific or Atlantic sea surface temperature (SST) drive circulation changes over the region that potentially lead to widespread drought. During the summer of 2014, Central America suffered major shortages in rainfall and one of the worst drought years in decades as a result of shifting SST in the tropical Pacific [6]. The Central American Dry Corridor is the most densely populated area of the Central American Isthmus and is subject to the greatest difference in precipitation between seasons [28]. Covariance between seasonal precipitation clearly shows the extent of the Central American Dry Corridor (Figure 2).

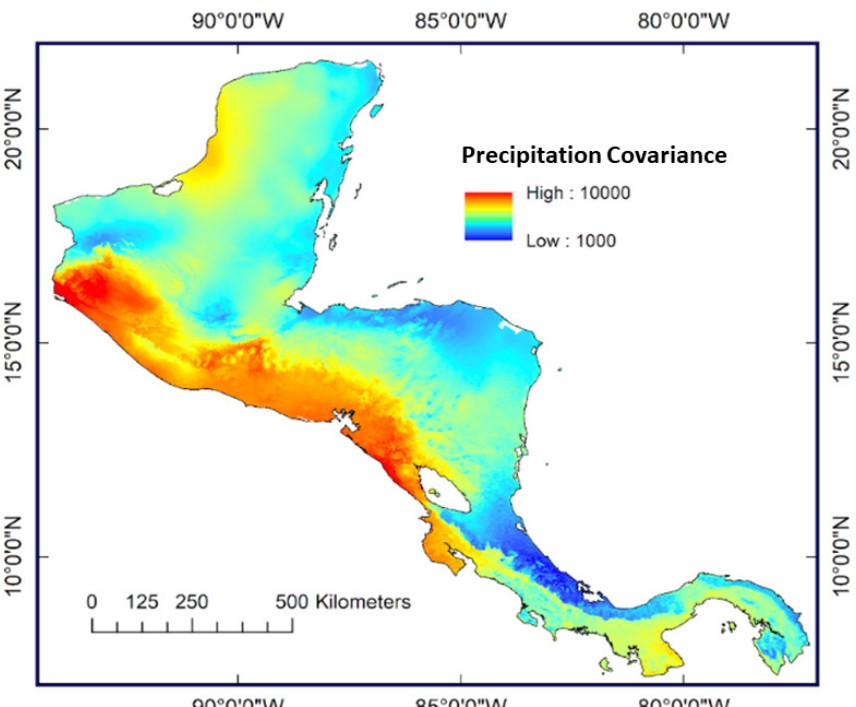

**Figure 2.** Covariance between seasonal precipitation (wettest quarter minus the driest quarter). Data from Global Surface Summary of the Day [29].

### *1.3. Fire and Burning*

Modern burning across the Central American Isthmus is driven by the complex interaction among climate, people and vegetation [30]. Climate-driven fire across the Central American Isthmus can be initiated by lightning strikes and/or prolonged periods of drought (e.g., [31]). The moist tropical forests of Central America rarely naturally burn, owing to the low frequency of lightning and high levels of moisture [30]. However, drier areas, such as dry tropical forests and savannas, experience more frequent natural fires due to the increased availability of seasonally desiccated biomass [32]. Coastal regions may also experience increased burning after a hurricane because of greater fuel loads and lightning strikes (e.g., [33]).

Burning within (i) previously unburned; (ii) once-burned; (iii) twice-burned; and (iv) more than twice-burned tropical forests (within a successional forest cycle) can lead to complex relationships and feedbacks associated with climate, fuel loads, and taxa persistence on temporal and spatial scales [34,35]. Fire in previously unburnt tropical

forests usually moves slowly along the ground with the intensity of a prescribed burn ~50 kW m$^2$ [36]. These fires primarily consume dry leaf litter; however, they cause mortality of around 95% to trees with thin tree bark [34,37]. As trees shed their fire-damaged biomass, the canopy of the forest opens (50%–70%), allowing for greater solar heating and air movement in the understory, drying the underlying forest fuels [35]. Fuel loads have been found to increase following an initial burn for up to 2 years after the initial event [34,38]. Previously burnt forests are more susceptible to future burning events, particularly during dry seasons, as a result of these increased fuel loads [34].

A secondary fire event in a once-burned forest is typically faster moving and more intense [35]. Cochrane [34] estimates that heat release in a once-burned tropical forest can reach 7500 kW m$^2$ and >7500 kW m$^2$ in subsequent burns [39]. Larger trees with thicker bark have little additional survival advantage during these more intensive burns, with mortality of up to 98% [40]. Fires in frequently impacted forests are substantially more severe in respect to intensity, flame height, penetration, residence time, and spread of burn. Recurring fires have the potential to remove trees from forested landscapes, leaving either scrub or grasslands for 20–80 years [17].

Humans can influence fire regimes in several ways, such as changing fuel availability and structure, e.g., removing biomass through the extraction of timber and deadwood, and controlling ignitions, lighting more or fewer fires regardless of season or weather [30]. Even today, people cannot completely control the fires they set, nor completely control fires caused by natural ignitions. For example, in April 2018, the Indio Maiz forest fire, started by local people, destroyed 5484 hectares of primary tropical moist broadleaf forests on the Moskito coast of Nicaragua [41,42]. The fire was thought to have been started by illegal settlement and poor agricultural practices.

*1.4. Population Growth and Land Use Change*

Population growth and land use change across Central America have accelerated over the past 50 years and are projected to continue expanding in the future (Figure 3). Increasing human populations are closely linked to the expansion of urban settlements, forest conversion for subsistence maize farming, and pasture creation [43,44].

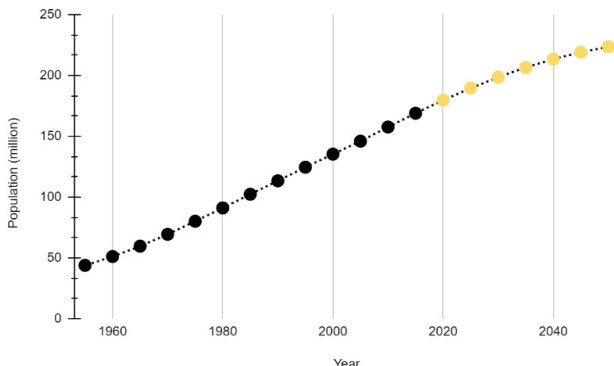

**Figure 3.** Population of Central America from 1950–2020 (black) and predicted population increase (yellow) through to 2050 [45].

Guatemala, Nicaragua, and Belize have been particularly impacted by increasing human populations, with 6866 km$^2$ of forest being converted into agricultural land between 2001 and 2010 [2]. From 1966 to 1994, Guatemala lost 22% of its forest cover to agricultural settlement, around 7% per decade [46]. Similar patterns are also noted across areas of the Yucatan Peninsula [2,47]. Across northern Guatemala, this deforestation is strongly associated with distance to human settlements and rapid population growth [48]. Nicaragua experienced the highest deforestation rate of any country in Central America, losing 7961 km$^2$ of forest from 2001–2010, with most of this land being converted into extensive pastures [2]. In contrast to the forest loss in Guatemala and Nicaragua during

the decade 2001–2010, Honduras and El Salvador experienced forest recovery, as 2335 km$^2$ of agricultural land was abandoned. However, whilst there is a considerable body of knowledge relating to the present-day situation in the Central American Isthmus, far less appears to be known about the trends over longer periods of time.

Palaeoecological records can be used to infer anthropogenic impacts on forests through the analysis of proxy data such as pollen and charcoal; however, it can be challenging to use such datasets to disentangle human from natural drivers of forest change. Anthropogenic impacts are typically inferred from the palynological record through: (i) the identification of known cultigens, such as *Capsicum* (peppers), Cucurbitaceae (gourds), *Maranta arundinacea* (arrowroot), *Phaseolus* (beans), and *Zea mays* (maize) [49]; (ii) the presence of "weedy taxa", such as Amarathaceae, Compositae, and *Polygonum* [50,51]; (iii) abrupt reductions in all or select arboreal taxa, such as *Quercus* (e.g., [50,52–54]); and (iv) increases in local and regional burning (e.g., [31,50,52–54]). It is often only through the combination of multiple lines of evidence that human impacts can be inferred (e.g., reduced forest taxa combined with an increase in known cultigens and an increase in fire). Across Central America, many of the domesticated taxa are native and occur in the wild as well as being cultivated by humans [49]. It can be difficult to determine if a taxon was native and grew naturally or was introduced and cultivated, and authors often simply report the most likely scenario (e.g., [50,52–56]).

The high biodiversity within the tropics presents significant challenges when attempting to identify anthropogenic cultigens due to the morphological similarities of some cultivated taxa to their native counterparts, e.g., Leguminosae spp.; however, some cultivated taxa such as *Zea mays* pollen are easily identifiable from other wild grasses and *Tripsacum* species due to their unique morphological properties including size (>65 um), a 2:3 annulus aperture, and greater wall thickness [57]. When *Zea mays* pollen is found in a palynological record, even in very low abundances, it suggests past human presence (e.g., [58]). In order to distinguish between anthropogenic and climate-driven burning, it is important to take a multi-proxy approach by using independent proxies to infer climate, vegetation, and fire (and, where possible, use multiple proxies to reconstruct fire) [59]. It is not possible to differentiate between anthropogenic and naturally occurring fires from charcoal records alone [31].

*1.5. Previous Reviews*

Previous summaries of vegetation dynamics through time across Central America have focussed upon the regional expansion of mesic forests in the Early–Middle Holocene (e.g., [60,61]) and the regional impacts of anthropogenic activities, particularly agriculture in the Maya lowlands, during the Pre-Classic 2000 B.C.E.–250 C.E., Classic 250–950 C.E., and Post-Classic 950–1500 C.E. Maya periods (e.g., [62–65]).

There are a number of review papers about past climate, which present syntheses of past precipitation for this region (e.g., [24,66–68]). These suggest spatial patterns of changes during select periods including (i) the onset into the Early Holocene 10000–7000 B.C.E.; (ii) the Middle Holocene 7000–2000 B.C.E.; (iii) the Medieval Climate Anomaly (MCA) 900–1250 C.E.; and (iv) the Little Ice Age (LIA) 1400–1850 C.E. It has been suggested that patterns of climatic forcing during the Late Holocene 2000 B.C.E. to the present are spatially and temporally heterogeneous, with signals increasingly masked by anthropogenic development after around 2000 B.C.E. [24]. Other reviews focus on periods of drought relating to the collapse of the Maya civilisation between 800 and 1100 C.E. (e.g., [67,68]). These review papers draw upon proxy evidence from both marine (e.g., the Cariaco Basin [69]) and terrestrial (e.g., [62]) archives in and around the Central American region.

No attempts to synthesise Holocene fire across the Central American Isthmus have been attempted. However, regional comparisons are discussed in detail (e.g., the Central Maya lowlands or highlands of Costa Rica), focusing upon the impacts of the Early–Mid Holocene (10000–6000 B.C.E.) climate (e.g., [70–72]) and Late Holocene (2000 B.C.E.–1500 C.E.) anthropogenic agrarian and architectural practices [31,59,72].

To date, there has been no attempt to systematically map palaeoecological proxies in this region. Research was therefore undertaken to create an evidence base of relevant proxies, and to organise the collected results in an accessible way for future use, including: palaeoenvironmental sciences, land use policy, and decision-making.

*1.6. Review Question*

Following good practice guidance for systematic reviews and systematic maps (CEE, 2018), the current review question is:

What is the palaeoenvironmental proxy evidence base for forests, water, and land use change across the Central American Isthmus spanning the Holocene?

## 2. Materials and Methods

Systematic evidence evaluations and synthesis methodologies are now widely used across many disciplines and have become a recognised standard for accessing, appraising and synthesising scientific information [73]. The need for rigour, objectivity, and transparency in reaching conclusions from a body of scientific information is evident in many areas of policy and practice, including questions relating to the environment. While systematic reviews and systematic maps have mostly been adopted for policy-relevant questions, the methods that dictate their conduct are of interest to academic literature reviews generally, given that transparency and repeatability are the cornerstones of science [74]. A systematic map was chosen for this review to capture the diverse range of evidence and methodological approaches prevalent in the palaeoenvironmental literature (Supplementary Material).

*2.1. Systematic Mapping*

Systematic review and systematic mapping follow the same rigorous processes to evaluate relevant evidence and minimise potential bias and lack of transparency, which are often found in traditional literature reviews (Table 1).

**Table 1.** Main differences between systematic evidence evaluations and traditional literature reviews.

| | Systematic Evidence Evaluation | Literature Review |
|---|---|---|
| Definition | High-level overview of primary research on a focused question that identifies, selects, synthesises, and appraises all high-quality research evidence relevant to that question. | Qualitatively summarises evidence on a topic using informal or subjective methods to collect and interpret studies. |
| Aim | Answers a focused question. Reduces bias. | Provide summary or overview of topic. |
| Question | Clearly defined and answerable question. Uses a recommended framework, e.g., PICO or PECO. | Can be a general topic or a specific question. |
| Components | Pre-specified eligibility criteria. Systematic search strategy. Assessment of the validity of findings. Interpretation and presentation of results. Discussion. Summary of key findings. | Introduction. Methods. Discussion. Conclusion. |
| Number of Authors | Two or more. | One or more. |
| Timeline | Months to years (average eighteen months). | Weeks to months. |
| Requirement | Thorough knowledge of the topic. Perform searches of all relevant databases. | Understanding of topic. Perform searches of one or more databases. |
| Value | Connects practicing scientists to high-quality evidence. Supports evidence-based practice. | Provides summary of literature on the topic. |

The evidence mapping process involves several stages: (i) comprehensive searching for evidence across a wide range of sources; (ii) selection of relevant evidence through screening and filtering; (iii) coding of key elements of individual studies in the evidence base; and (iv) presentation of the evidence base [73].

### 2.2. PECO Framework

In accordance with systematic evidence evaluation practice [73], key elements of the question were defined into four categories: (i) population; (ii) exposure; (iii) counterfactual; and (iv) outcome (Table 2).

**Table 2.** Population, Exposure, Counterfactual, Outcome (PECO) framework.

| PECO | Definition | Description |
|---|---|---|
| Population | Populations of subject(s) of relevance to the review question. | Terrestrial palaeoenvironmental proxies interpreted to reconstruct forests, water, or land use change during the Holocene and within the Central American Isthmus. |
| Exposure | Environmental variable impacting the populations or to which the subject populations are exposed. | A window of time. |
| Counterfactual | What the exposure is compared to. Either a control with no exposure or an alternative counterfactual scenario. | A different window of time. |
| Outcome | Consequences of the exposure. All relevant variables that can be reliably measured. | Instances of change relating to forests, water, or land use change. |

### 2.3. Literature Search Strategy

The search strategy followed guidance outlined in Livoreil et al. [75]. Four key word search strings were refined with iterative testing using three relevant online bibliographic databases: Web of Science core collection (https://apps.webofknowledge.com/), Scopus (https://www.scopus.com/) and CAB Abstracts (https://www.cabdirect.org/) (accessed on 26 April 2018) (Table 3). These search strings were composed of key words defining the geographic location, type of archive, time period, and palaeoenvironmental proxies (Appendix A). Key words were combined using Boolean operators to search the online bibliography databases. The geographic location was constrained to terrestrial evidence situated between the Isthmus of Tehuantepec and the Isthmus of Panama. Database searching took place on the 10 August 2020 in both English and Spanish. Language limitations were not applied to any database, allowing for all papers which had an English or Spanish title, abstract or keywords to be found.

**Table 3.** Database search.

| Database | Type of Literature | Description of Search |
|---|---|---|
| Web of Science | Platform of bibliographic databases | University of Oxford Core Collection of databases, 1945–2020. |
| Scopus | Bibliographic database | All records to 2020. |
| Cab Abstracts | Bibliographic database | All records included in the 1910–2020 databases. |

### 2.4. Comprehensiveness of Search

The four search strings were iteratively tested using a Test Library of papers known to be of central relevance to the current review (Appendix B) to ensure that the search was comprehensive but not too broad [75]. Search strings were refined several times to remove terms containing generic nomenclature while still maintaining all relevant papers.

Additional search terms were added to capture papers from the test set that did not specify a period of time (e.g., [76]), or were too specific in their period of time (e.g., [33]). Search terms were iteratively modified and optimised until the entire Test Library was returned.

### 2.5. Inclusion Criteria

Inclusion criteria were developed and applied to each article retrieved from all of the sources listed in Table 2. An article was selected if it satisfied all of the following criteria (Table 4):

**Table 4.** Inclusion criteria.

| PECO | Inclusion Criteria |
|---|---|
| Population | All studies that present primary palaeoenvironmental proxy data collected from a terrestrial environmental archive situated within the Central American Isthmus (terrestrial records only). Studies must represent a period of time within the Holocene. |
| Exposure | The environmental proxy data must be representative of a point in time. |
| Counterfactual | The environmental proxy data must be representative of a different point in time. |
| Outcomes | Palaeoenvironmental proxy data must have been interpreted to represent an environmental variable. Studies reconstruct an aspect of the past environment relating to forests, and water or land use change. |

Articles that failed to meet the inclusion criteria were excluded from the review. Records that presented only syntheses, reviews, remote sensing data, or models were also excluded. There was no limitation for language regarding the inclusion of articles. There was no limitation of date regarding the publication of articles. Articles may contain more than one study, spread across time or space, or present different outcomes. Each independent outcome was termed "study". Studies may be linked in separately published articles, and linkages were recorded where possible. Individual studies (whether from the same article or not) were given separate unique identifiers. Articles were excluded if they did not contain data of relevance to the outcomes (above). Discussion papers or thought pieces will fall into this category. These papers can be noted and used for a better understanding of the background and context to the review questions, but will not form part of the evidence base.

### 2.6. Article Screening

Following good practice guidance [77], screening was conducted in two stages. Firstly, simultaneous title and abstract screening was carried out using Abstrackr software [78]. Abstrackr is a free online tool into which a formatted citation list is uploaded.

The list can be assessed for abstract screening by several reviewers. Each paper was screened by two reviewers to reduce bias. Reviewers individually appraised studies against the inclusion criteria. Whenever there was doubt about inclusion, reviewers discussed with each other in order to reach a common understanding of the application of the inclusion criteria. Cohen's kappa was applied to assess reviewer agreement [79]. Cohen's kappa is a quantitative measure of reliability for two raters who are rating the same thing, corrected for how often that the raters may agree by chance. A kappa of 0.6277 was established between reviewers on a test sample of 100 studies, which showed good agreement between reviewers and therefore confidence that the inclusion criteria were clear enough to identify all the relevant papers in the searched databases. The second stage was full-text screening. Full texts that met the eligibility criteria for inclusion were then reviewed for coding and data extraction.

### 2.7. Article Coding and Data Extraction

A detailed coding sheet was devised to code the studies that met inclusion criteria at full-text documenting the article metadata, study metadata, study proxy, cross cutting lens, time data, presentation of data, and availability of original, raw data (Table 5).

**Table 5.** Data coding and extraction for studies that met the inclusion criteria at full-text.

| Coded Elements | Description |
|---|---|
| Article metadata | Authors, Title, Journal, Year of publication, Abstract, etc. |
| Study area metadata | Information about the study site including precise location. |
| Study proxy | Information about the study proxy, including data collection and interpretation. |
| Cross Cutting Lens | Study evidence for forests, and land use change or water. |
| Time data | Information about the time period examined, dating methods, materials, number of dates, constrained sequences, calibration, and reported errors. |
| Presentation of data | Temporal or depositional presentation of proxy data |
| Availability of original data | Accountability of the presented data. Are the graphically presented data available with the published article? |

### 2.8. Study Quality

Metadata documenting the reproducibility of the study were extracted to give an indication of the quality of the research design; however, in common with most systematic evidence maps, critical appraisal of all studies selected at full-text was not undertaken and no studies were excluded from the evidence base on the basis of likelihood of bias [73].

## 3. Results

### 3.1. Selection of Articles

The systematic literature search returned 12,474 publications, of which 3508 were removed as duplicates. There were 8972 articles screened, from which 425 were reviewed at full-text level. Individual study sites and proxies were recorded, separating articles with multiple study sites and/or proxies from each publication. Proxy data were extracted for 149 publications, which presented 167 study sites and 648 proxy records, of which 539 proxy records from 160 sites presented evidence for forests, water, or land use change (Figure 4). There were 109 proxy datasets that did not present evidence for forests, water, or land use change. Only studies (proxy datasets) that interpreted proxy data for forests, water, or land use change were assessed in the evidence base.

### 3.2. Interactive Systematic Map

Using the geo-locations of all included studies, results of the evidence base are presented on an interactive map. The map is available at: https://oxsrev.github.io/evidencemaps/palaeo_2021/ (accessed on 26 April 2018).

Operational details can be found on the website. The geographic map of the evidence base was constructed using the Open-Source Thalloo mapping software [80]. Each circle on the map represents one location at which an evidence point was generated, or a regional cluster if more than one location occurred within 50 km. Circle size represents the count of evidence points that occurred at the location. Pie segments represent a percentage of the evidence points at a location for each study.

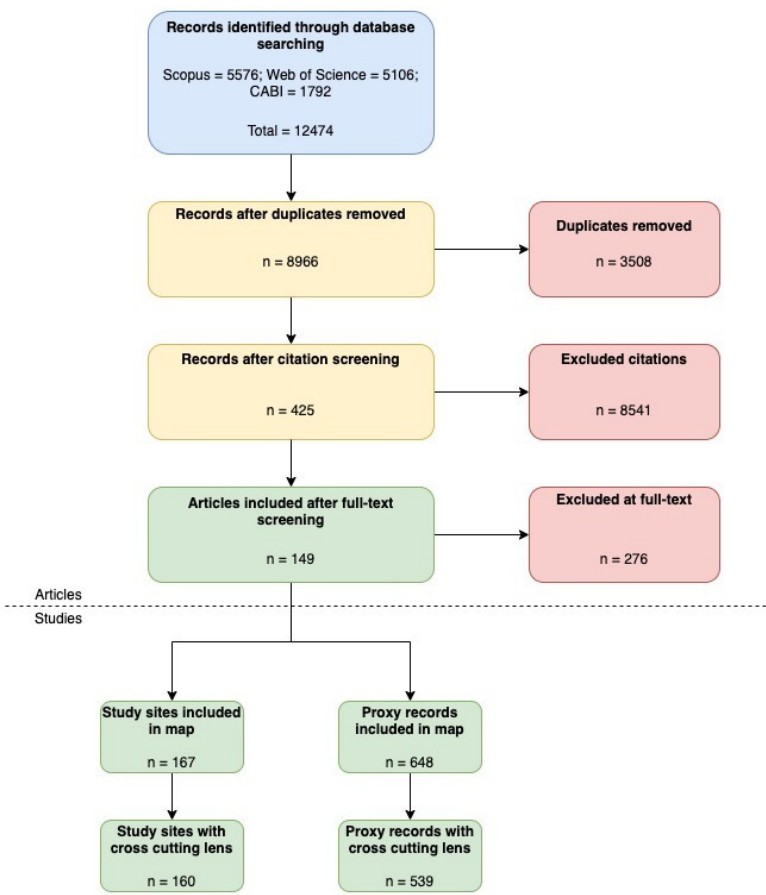

**Figure 4.** Reporting standards for Systematic Evidence Syntheses (ROSES) diagram detailing the steps taken in identifying literature, screening articles, and splitting individual studies from articles.

### 3.3. Source of Studies

The number of studies has increased over the past four decades, with the earliest studies published in 1979 C.E. (Figure 5A). There has been a particular increase in the number of studies published in the last decade ($n = 272$, 49.5%). The most popular journals for publishing these studies are the *Journal of Paleolimnology* ($n = 78$, 14.5%), *Quaternary Research* ($n = 55$, 10.2%), and *Quaternary Science Reviews* ($n = 38$, 7.1%) (Figure 5B). The majority of studies were published in English ($n = 535$, 99.3%) and a few in Spanish ($n = 4$, 0.7%) (Figure 5C).

### 3.4. Type of Environmental Archive

There were 13 types of environmental archives from which the palaeoenvironmental data were collected (Figure 6). Most studies extracted their data from lake sediments ($n = 360$, 66.8%), swamp sediments ($n = 35$, 6.5%), and speleothems ($n = 23$, 4.3%). Proxies from speleothems were exclusively used to reconstruct climate.

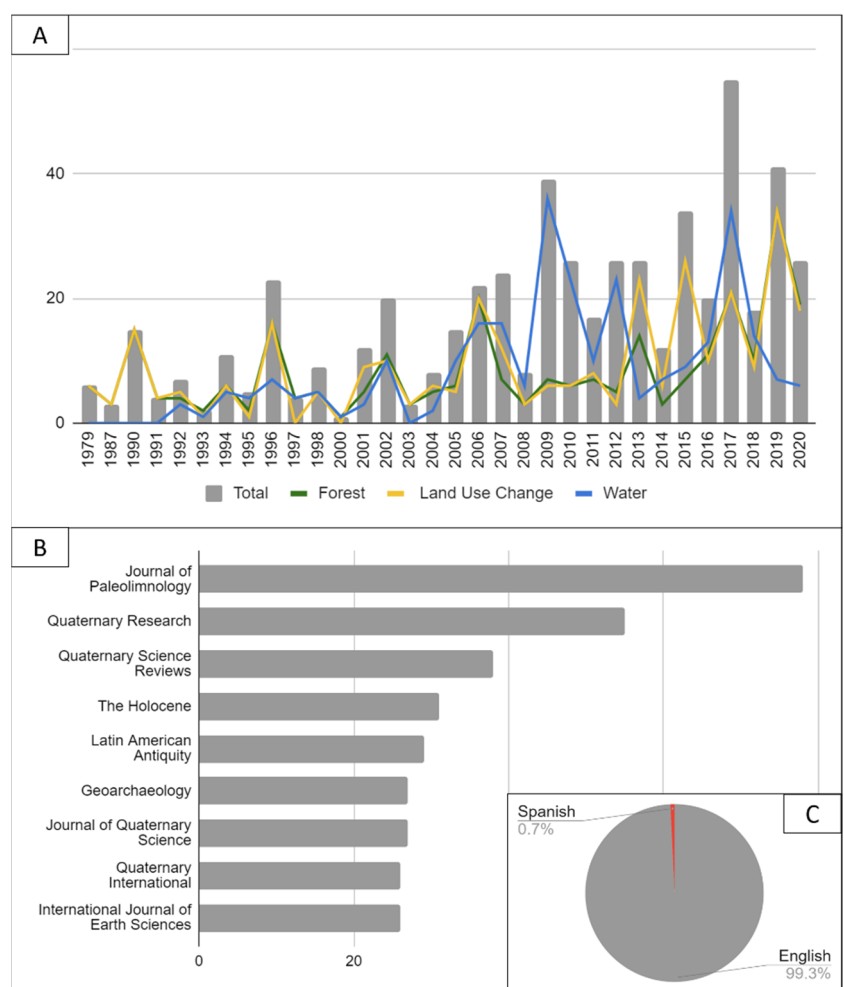

**Figure 5.** Source of studies presented in the evidence base: (i) Total; (ii) Forest; (iii) Land Use Change; (iv) Water. (**A**) Number of studies by year of publication. (**B**) Number of studies by Journal; (**C**) Published Language.

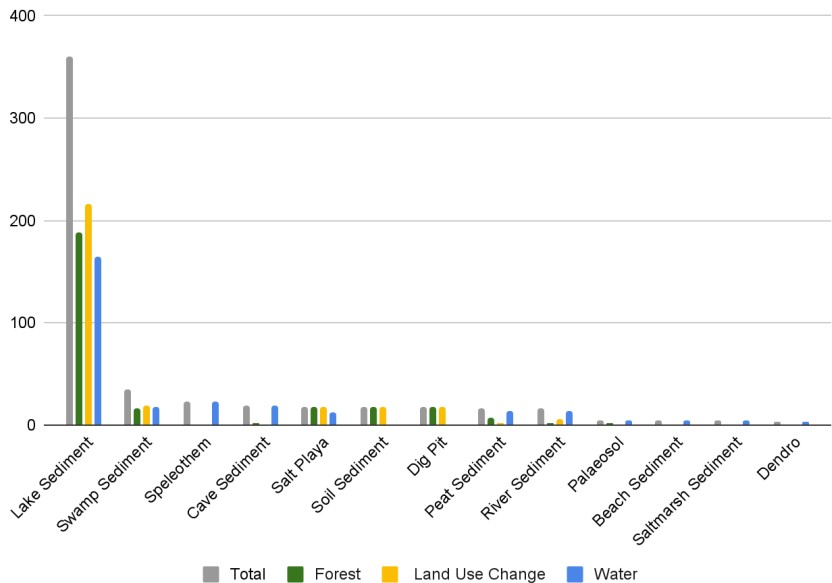

**Figure 6.** Number of studies extracted from different environmental archives.

### 3.5. Location of Studies

Studies have a geographic focus on Guatemala (*n* = 175, 32.5%), Belize (*n* = 112, 20.1%), and Mexico (*n* = 95, 17.6%) (Figure 7A). Specifically lowland areas (<1000 m.a.s.l.) of Guatemala (149, 27.6%), Belize (112, 20.8%), and Mexico (93, 17.3%) comprise a total of 354 (65.7%) studies (Figure 7B). Topographically, 452 (83.9%) records were collected from lowland sites and 87 (16.1%) from upland sites (Figure 7B,C).

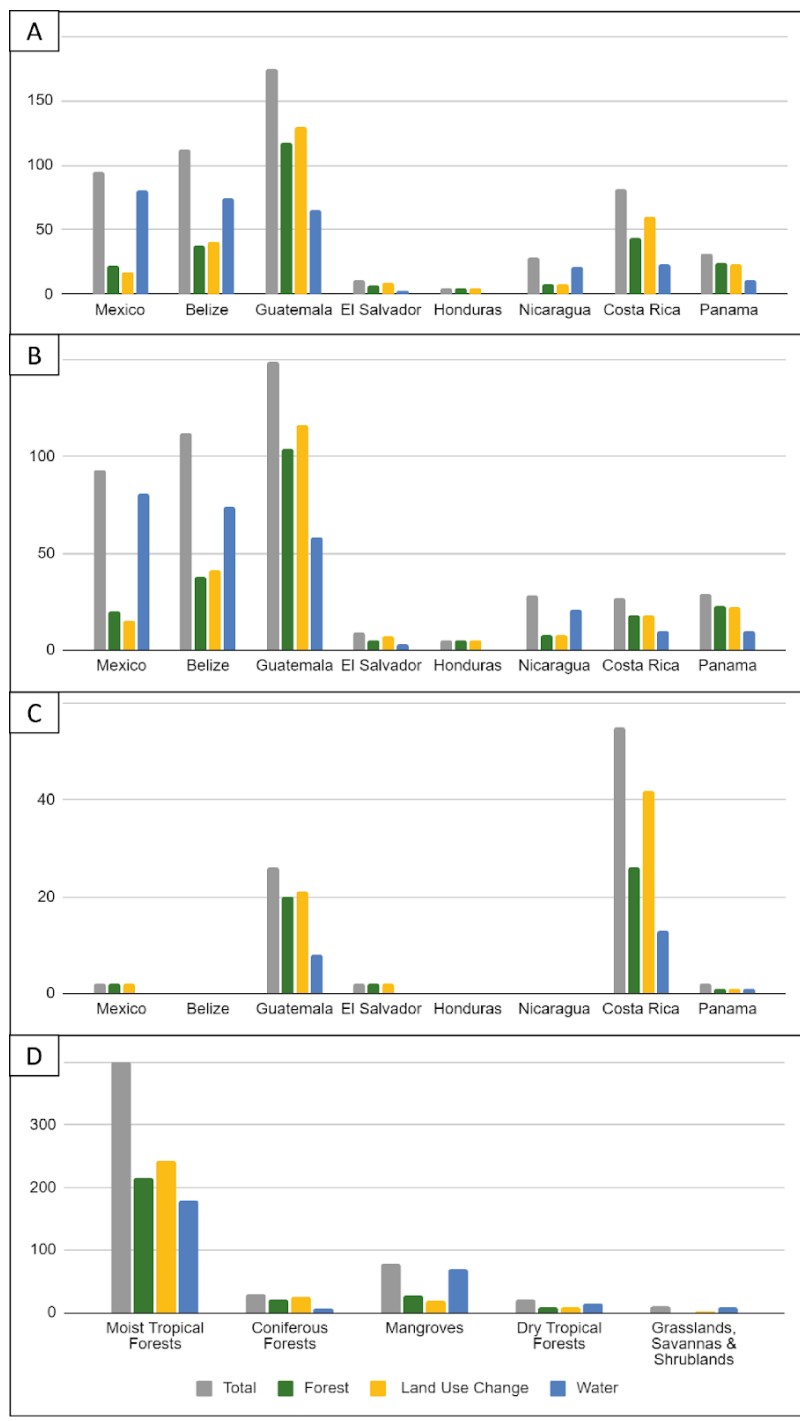

**Figure 7.** Location of studies presented in the evidence base: (i) Total; (ii) Forest; (iii) Land Use Change; (iv) Water. (**A**) Number of studies in each country; (**B**) Number of studies in each country >1000 m.a.s.l.; (**C**) Number of studies in each country >1000 m.a.s.l.; (**D**) Number of studies in each forest type.

Spatially, the largest data gaps are records from the highland areas of Honduras ($n = 0$, 0%), Nicaragua ($n = 0$, 0%), Belize ($n = 0$, 0%), Mexico ($n = 2$, 0.4%), Panamá ($n = 2$, 0.4%), El Salvador ($n = 2$, 0.4%), and Guatemala ($n = 26$, 4.8%); and lowlands of Honduras ($n = 5$, 0.9%), El Salvador ($n = 9$, 1.7%), Costa Rica ($n = 27$, 5%), Nicaragua ($n = 28$, 5.2%), and Panamá ($n = 29$, 5.3%). A total of only 130 (24.1%) studies have been published from these regions collectively.

Studies from moist tropical forests have been the most prominent ($n = 400$, 74.2%), with relatively few produced from mangroves ($n = 78$, 14.5%), coniferous forests ($n = 29$, 5.4%), dry tropical forests ($n = 21$, 3.9%), and grassland areas ($n = 11$, 2%) (Figure 7D).

*3.6. Temporal Coverage*

Temporal coverage of the evidence base denotes the lowest temporal coverage between 10000 and 9000 B.C.E. ($n = 39$, 7.2%) and the highest cover from 0–1950 C.E. ($n = 332$, 61.6%). The number of palaeoenvironmental records available increases from the Early Holocene through to the present day (Figure 8); however, dating methods and practices vary across studies. Only 122 (22.5%) studies constrain their depositional sequence (date the top and the bottom of their sequence), with 20 (3.7%) publishing their raw palaeoenvironmental study data and 102 (18.8%) not publishing their raw palaeoenvironmental study data. These studies are considered low risk in terms of good methodological practice for sequence dating (well-constrained). Medium-risk studies ($n = 212$, 39.1%) have dated the bottom of their sequence only and do not report a hiatus in their record, or they have constrained any hiatus with independent dates (moderately constrained). High-risk studies ($n = 205$, 36.4%) are studies that have not dated the bottom of their sequence (poorly constrained) (Figure 9A). Not dating the bottom of a sequence is particularly problematic for studies that have been presented in their publications as measurements of depth only ($n = 317$; 58.9%) when they are discussed as time (Figure 9B), and this is further complicated when the raw study data are not available with the published article ($n = 496$, 92.2%) (Figure 9C).

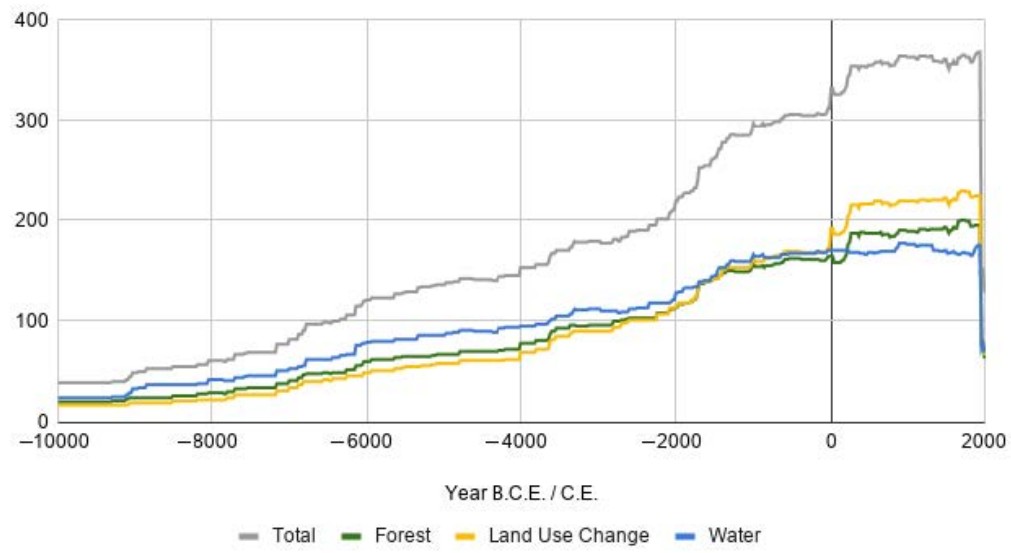

**Figure 8.** Number of studies and temporal coverage of the Holocene in 20-year time slices.

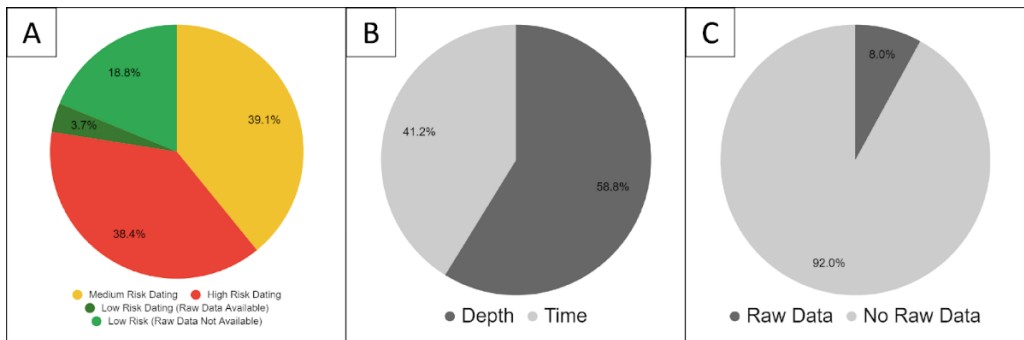

**Figure 9.** (**A**) Percentage of studies with a high (poorly constrained sequence), medium (moderately constrained sequence) or low (well-constrained sequence) risk when compiling the age–depth model; (**B**) Percentage of studies presented as depth or time; (**C**) Percentage of studies publishing their raw data with the journal article.

### 3.7. Environmental Proxies

There were 17 proxies used to indicate forests, water (hydroclimate, sea level, or tropical storms), or land use change (including agriculture) from the studies included in the evidence base (Table 6).

**Table 6.** Analyzed variable in paleoenvironmental archives and the environmental variable(s) for which it served as a proxy.

|  | Forests | Hydro Climate | Sea Level | Tropical Storms | Agriculture | Land Use Change |
|---|---|---|---|---|---|---|
| Charcoal (Macroscopic) |  |  |  |  | X | X |
| Charcoal (Microscopic) |  |  |  |  | X | X |
| Diatoms |  |  |  | X |  | X |
| Foraminifera |  | X | X |  |  |  |
| Geochemistry |  | X |  |  |  | X |
| Grain Size Analysis |  | X | X |  |  |  |
| Inorganic Carbon |  | X | X | X | X | X |
| Magnetic Susceptibility |  |  | X | X |  | X |
| Organic Matter |  | X | X | X | X | X |
| Ostracoda |  |  | X |  |  | X |
| Phytoliths | X |  |  |  | X | X |
| Pollen | X | X | X | X | X | X |
| Stratigraphic Description |  |  | X | X |  | X |
| Testate Amoebae |  | X | X |  |  |  |
| $\delta^{13}$C | X | X | X | X | X | X |
| $\delta^{15}$N | X | X | X |  | X | X |
| $\delta^{18}$O |  | X | X | X |  |  |

### 3.8. Forest Evidence

The evidence base for forest dynamics was reported across 64 sites presenting 272 (49.5%) proxy datasets: 85 (31.3%) pollen; 16 (5.9%) phytolith; 27 (9.9%) $\delta^{13}$C; 6 (2.2%) $\delta^{15}$N (Figure 10). These records are concentrated in the lowlands ($n = 221$, 81.3%) of Guatemala (104, 38.2%), Belize ($n = 38$, 14%), Mexico (20, 7.3%), and Panama ($n = 23$, 8.5%). Studies have primarily been conducted in moist tropical forests ($n = 215$, 79%). Relatively few studies have come from Honduras ($n = 5$, 1.8%), El Salvador ($n = 7$, 2.6%), and Nicaragua ($n = 8$, 2.9%), and are also largely absent from the highland regions for all countries ($n = 27$, 9.9%). Studies from dry tropical forests ($n = 21$, 3.9%), coniferous forests ($n = 29$, 5.4%), and grassland ($n = 11$, 2%) biomes are also limited in number. Just over one quarter of the forest evidence base ($n = 72$, 26.5%) studies are located within the Central American Dry Corridor. Changes to forest cover, dynamics, and composition are linked to water as

driven by: (i) hydroclimate, sea level, and tropical storms; or (ii) land use change relating to human activities including land clearance for settlement, the extraction of building materials, and agrarian practices.

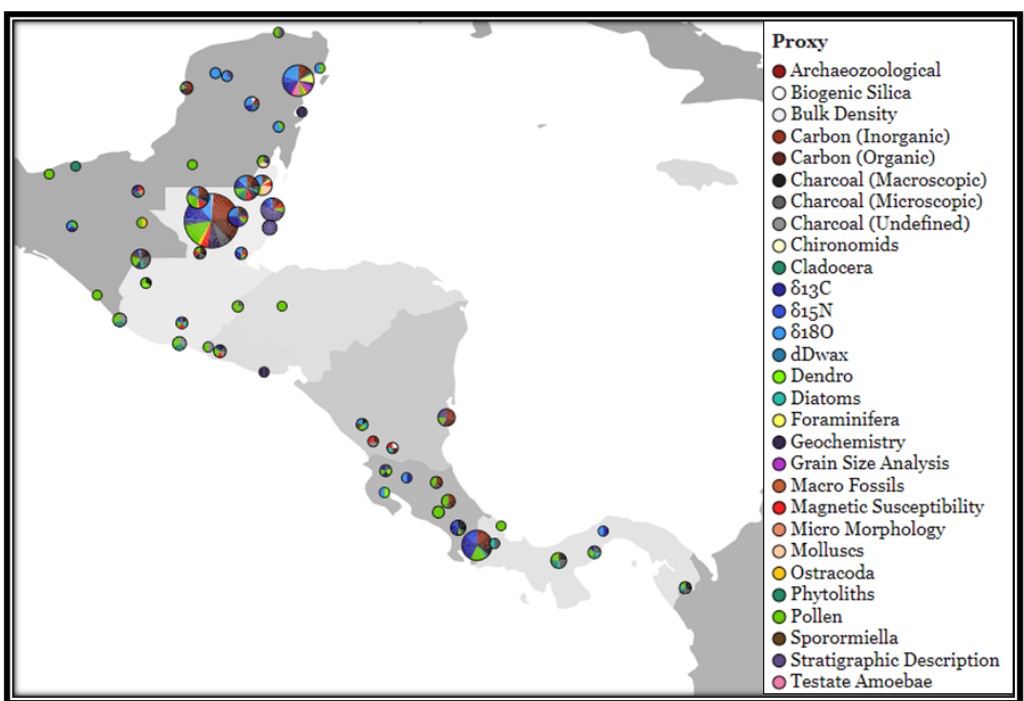

**Figure 10.** Location and number of studies presenting proxy evidence for: forests, water, and land use change: https://oxsrev.github.io/evidencemaps/palaeo_2021/ (accessed on 26 April 2018).

### 3.9. Water Evidence

The evidence base for studies reporting on water is from 43 sites presenting $n = 279$ studies that examine changes in hydroclimate ($n = 225$, 80.7%), sea level (40, 14.3%), and the impacts of tropical storms (41, 14.7%) (Figure 10). The main proxies used to infer these changes include: $\delta^{18}O$ ($n = 54$, 19.4%); pollen ($n = 39$, 14%); stratigraphic description ($n = 31$, 11.1%); $\delta^{13}C$ ($n = 28$, 10%); and inorganic carbon ($n = 19$, 6.8%). The main types of depositional archives preserving a water signal include: lake sediments ($n = 165$, 59.1%); speleothems ($n = 23$, 8.2%); cave sediments ($n = 19$, 6.8%); and swamp sediments ($n = 17$, 6.1%). Spatially, most records published are from lowland areas and are predominantly clustered on: the Yucatan Peninsula and Quintana Roo, Mexico ($n = 73$, 33.3%); Belize ($n = 74$, 33.8%); and the Petén district, Guatemala ($n = 46$, 21%). Relatively few studies have been produced from the highland regions ($n = 22$, 10%), limited to: Costa Rica ($n = 13$, 5.9%); Guatemala ($n = 8$, 3.7%); and Panama ($n = 1$, 0.5%). Similarly, in the lowlands, few studies have been published from Honduras ($n = 0$, 0%); El Salvador ($n = 3$, 1.4%); Costa Rica ($n = 10$, 4.6%); and Panama ($n = 10$, 4.6%). Most hydroclimate studies have been published from areas within moist tropical forests ($n = 179$, 64.2%), whereas studies examining sea level are primarily from moist tropical forests ($n = 25$, 59.5%) and mangrove sites ($n = 17$, 40.5%). Evidence for tropical storms is also primarily reported from mangrove sites ($n = 28$, 68.3%) and moist tropical forests ($n = 8$, 19.5%).

### 3.10. Land Use Change Evidence

Land use change is reported across 72 sites presenting $n = 300$ studies that examine the impacts of deforestation ($n = 293$, 97.7%) and agrarian practices ($n = 240$, 80%) (Figure 10). The dominant types of proxies used in studies of land use change are: pollen ($n = 72$, 24%); charcoal ($n = 49$, 16.3%); $\delta^{13}C$ ($n = 34$, 11.3%); organic carbon ($n = 36$, 12%); and inorganic carbon ($n = 30$, 10%). Studies for land use change are prevalent across the lowlands of

Guatemala ($n$ = 137, 45.7%), Belize (41, 13.7%), and highlands of Costa Rica ($n$ = 42, 14%). Evidence for land use change has predominantly come from lake sediments ($n$ = 217, 72.3%); swamp sediments ($n$ = 19, 6.3%); dig pits ($n$ = 18, 6%); soil sediments ($n$ = 18, 6%); and salt playas ($n$ = 18, 6%). Relatively few studies have been published from Honduras ($n$ = 5, 1.7%); Nicaragua ($n$ = 8, 2.7%); and El Salvador ($n$ = 9, 3%). Most land use change has been documented from studies within moist tropical forests ($n$ = 243, 81%), with only a fraction recording land use change in dry tropical forests ($n$ = 8, 2.7%) and grasslands ($n$ = 3, 1%).

## 4. Discussion

### 4.1. Knowledge Gaps in the Evidence Base

There has been a particular increase in the number of studies published in the last decade (49.5%). Most studies extracted their data from lake sediments (66.8%). Spatially, studies have a geographic focus on Guatemala (32.5%), Belize (20.1%), and Mexico (17.6%), and have been conducted in lowland sites (83.9%) in moist tropical forests (74.2%). The most studied period of time covers the last two millennia: 0–1950 C.E. (61.6%). Evidence for forests is primarily inferred from pollen, phytolith, and $\delta^{13}$C proxy data (37.1%). Evidence for water is primarily inferred from $\delta^{18}$O, pollen, and stratigraphic descriptions (44.5%). Evidence from which land use change is primarily inferred comes from pollen, charcoal, $\delta^{13}$C, and organic/inorganic matter (73.6%).

Spatially, the largest data gaps in the evidence base are located in: (i) the highlands of Honduras (0.0%), Nicaragua (0.0%), Belize (0.0%), Mexico (0.4%), Panamá (0.4%), El Salvador (0.4%), and Guatemala (4.8%); and (ii) the lowlands of Honduras (0.9%), El Salvador (1.7%), Costa Rica (5%), Nicaragua (5.2%), and Panamá (5.3%). Limited studies have also been published from grassland areas (2%), dry tropical forests (3.9%), coniferous forests (5.4%), and mangroves (14.5%). Temporal coverage of the evidence base denotes the lowest temporal coverage between 10000 and 9000 B.C.E. at the onset of the Holocene (7.2%).

Few studies published their raw data with their articles ($n$ = 20, 3.7%), which can be problematic if trying to obtain the data for meta-analysis. Authors may have deposited their data in other repositories such as: Neotoma (https://www.neotomadb.org/); National Centers for Environmental Information (https://www.ncei.noaa.gov/); the International Tree Ring Databank (https://www.ncdc.noaa.gov/data-access/paleoclimatology-data/datasets/tree-ring); Paleobiology Database (https://paleobiodb.org/); the Paleoclimatology data holdings at the National Oceanic and Atmospheric Administration (https://www.ncdc.noaa.gov/data-access/paleoclimatology-data); Pangaea (https://www.pangaea.de/); MorphoBank (https://morphobank.org/); the Interdisciplinary Earth Data Alliance (https://www.iedadata.org/); the Strategic Environmental Archaeology Database (https://www.sead.se/); and the Limnological Research Center and Continental Scientific Drilling Office (http://lrc.geo.umn.edu/laccore/repository.html) (accessed on 26 April 2018). Alternatively, authors may provide their data if requested via email or shared on a research sharing platform such as Researchgate (https://www.researchgate.net/). For this systematic map, authors were not contacted for their raw data, and they may have deposited their raw data in publicly accessible repositories. Therefore, it is to be acknowledged that these data may exist and be available if searched for or requested; however, the absence of these raw, original data with the published article presents a worrying barrier to future access, use, and quality assessment of these datasets.

### 4.2. Future Analyses and Research

The systematic map of the evidence base highlights six key review topic areas that could be targeted, if the raw data could be obtained.

### 4.2.1. Dating Uncertainty and Standardising Reporting

The establishment of guidelines and quality of dating practices for palaeoenvironmental research across the Central American Isthmus is required. Accurately dating palaeoenvironmental records presents numerous challenges and uncertainty. This is due to a number of factors including poor methodological practices, but also: (i) the need to re-evaluate dates, specifically radiocarbon, to ensure suitable material was originally selected for dating; (ii) the use of un-calibrated and uncorrected dates (i.e., infiltration of allochthonous carbon or impacted by the hard water effect); (iii) the presentation of data against depth instead of time; and (iv) differences in the methods used for interpolation or modelling of ages when creating an age–depth model (e.g., Bayesian, multiple regression, or linear approaches). These uncertainties would need to be carefully considered when interpreting and comparing both individual and multiple records, and to enable quantitative meta-analysis to be successfully undertaken. In order to quantify and evaluate the time uncertainty aspects of palaeoenvironmental datasets, the age–depth model and proxy measurement data should be standardised, and the palaeoenvironmental data should be re-evaluated against the calculated time uncertainty.

### 4.2.2. Land Use Change across Space and Time

As populations expanded during the Pre-Classic (2000 B.C.E.–250 C.E.) period, state-level societies emerged in the Northern Isthmus, which led to sedentary habitation and the demand for increased space and resources. Identifying the impacts of different land uses across space and time would contribute towards our understanding of the long-term impacts on forest community composition and arboreal cover. Insights into the impacts of horticulture, slash and burn agriculture, timber extraction for building and burning, land clearance, ranching, and mining can be evaluated spanning the Holocene, e.g., impacts from early nomadic or semi-nomadic hunter gatherers, Pre-Columbian state-level societies, European contact, Latin American Independence, and the present day.

### 4.2.3. Dispersal Pathways of Agriculture

Early agricultural practices have been documented both to the north and to the south of the Central American Isthmus; however, the role and influence of agriculture across the Central American Isthmus are currently poorly understood. Through the review and synthesis of all identified proxy environmental evidence for Holocene-period plant exploitation and cultivation practices across the Central American Isthmus, we can gain an understanding of: (i) how and when agricultural practices entered the Central American Isthmus; (ii) what pathways across the Central American Isthmus agriculture took; and (iii) what plant taxa were being used for agriculture and agrarian practices.

### 4.2.4. The Role and Impacts of Fire and Burning

The timing of widespread climatically driven fire and spatially heterogeneous anthropogenic burning is not well understood or quantified on a regional scale for the Central American Isthmus. Most records have been collected from lowland tropical moist broadleaf forests in lowland Mexico and Guatemala. Fire is likely to have played an important role in driving vegetation diversity and structure, particularly where human populations were dense, such as during the peak of the Classic period (250–900 C.E.) with Maya city-states. A review examining the drivers and timing of natural fire or anthropogenic burning across the Central American Isthmus within different forest types would improve our understanding of the role of fire in these forests through time.

### 4.2.5. Changes in Hydro-Climate, Water Availability, and the Risk of Tropical Storms

The homogeneous and heterogeneous patterns of precipitation across the Central American Isthmus spanning the Holocene are poorly understood due to spatially and temporally limited records. Hydroclimatic changes are usually described with the terms "wetter" and "drier", which refer to the change in the quantity and/or presence of water.

In order to make informed predictions on the impacts of climate change, it is essential to be able to compare across temporal and spatial scales [81]. Standardising proxy data (e.g., $\delta^{18}O$) and modelling them against modern measurements would enable a quantifiable understanding of past hydro-climatic changes for specific locations.

### 4.2.6. Forest Resilience and Recovery

Identifying the recovery and resilience of vegetation after a disturbance event such as natural hazards (climate change, tropical storms, fire) and agricultural activities is integral to our understanding of different types of ecosystem resilience and the factors that help contribute towards increasing the resilience of a given ecosystem. Using the palaeoecological records identified in this study, disturbance events and rates of recovery could be quantitatively identified and analysed from high-resolution datasets using locally defined thresholds (e.g., [56]). This would enable measures to be taken for conservation and climate change management adaptation strategies in vulnerable biomes. Methodological approaches taken in Adolf et al. [82] or Cole et al. [83] could be considered to address this knowledge gap.

Meta-analyses of these identified knowledge gaps would enable robust long-term insights to inform on land use policy and decision-making, as a response to (a) climate change, (b) increasing population pressures, and (c) land use change. Integrating modern measurements, historical records, and palaeoenvironmental data would enable quantifiable estimates of impacts and provide the knowledge base to make informed policy decisions in the future.

### 4.3. Limitations of This Review

Despite our efforts to be inclusive with search terms and languages, some important studies were undoubtedly missed, particularly those recorded in the "grey literature". This is particularly true of "inhouse" publications from local institutions and academics not involved in the international community. Spanish literature is notably absent and warrants further investigation into identifying repositories. With the exception of CAB Abstracts, which includes non-journal articles (e.g., books, book chapters, reports), these databases index little "grey literature" or "dark data," which are abundant but difficult to access [84,85]. The bibliographic databases searched included a wide range of academic journals across the sciences, social sciences, and humanities; these three databases did not index all potentially relevant journals, including *World Archaeology*, *Annual Review of Anthropology*, and *Ancient Mesoamerica*.

Databases searched only index literature published since 1910 C.E. (CAB Abstracts), 1945 C.E. (Web of Science), and 1990 (Scopus). Implication for Policy and Research; older articles before these dates would need to be hand-searched in a physical library or archive. This has cost implications, and the need for comprehensiveness has to be balanced against time, as is the case for requesting datasets from authors by email or other contact.

**Supplementary Materials:** The following are available online at https://www.mdpi.com/article/10.3390/f12081057/s1, palaeoenvironmental proxy evidence base data.

**Author Contributions:** Conceptualization, W.J.H., G.P., N.S., S.N. and K.J.W.; Data curation, W.J.H. and L.P.; Formal analysis, W.J.H.; Funding acquisition, W.J.H. and N.S.; Investigation, W.J.H.; Methodology, W.J.H. and G.P.; Supervision, G.P., N.S., S.N. and K.J.W.; Validation, G.P., S.N. and K.J.W.; Visualization, W.J.H. and L.P.; Writing—original draft, W.J.H. and K.J.W.; Writing—review and editing, W.J.H., G.P., N.S., S.N., L.P. and K.J.W. All authors have read and agreed to the published version of the manuscript.

**Funding:** This research was funded by the Natural Environment Research Council of the United Kingdom NE/L002612/1, US National Science Foundation EAR-1502989, St Edmund Hall, University of Oxford, and Oxford Systematic Reviews LLP.

**Acknowledgments:** The authors would like to thank the two reviewers, Mark Brenner and anonymous, for their comments and recommendations for improving this article.

**Conflicts of Interest:** The authors declare no conflict of interest.

## Appendix A. Search Terms

("Central America" OR centroamerica OR mesoamerica OR mexico OR mejico OR tabasco OR chipas OR campeche OR yucatan OR "Quintana Roo" OR belize OR belice OR cayo OR corozal OR "Orange Walk" OR "Stann Creek" OR toledo OR guatemala OR "Alta Verapaz" OR "Baja Verapaz" OR chimaltenango OR chiquimula OR "El Peten" OR "El Progreso" OR "El Quiche" OR escuintla OR huehuetenango OR izabal OR jalapa OR jutiapa OR quetzaltenango OR retalhuleu OR sacatepequez OR "San Marcos" OR "Santa Rosa" OR solola OR suchitepequez OR totonicapan OR zacapa OR "El Salvador" OR "La Union" OR "San Miguel" OR "La Libertad" OR "Santa Ana" OR usulutan OR ahuachapan OR chalatenango OR sonsonate OR morazan OR "La Paz" OR "San Salvador" OR cuscatlan OR cabanas OR "San Vicente" OR honduras OR francisco OR morazan OR choluteca OR "El Paraiso" OR lempira OR olancho OR "Santa Barbara" OR "La Paz" OR copan OR valle OR comayagua OR yoro OR intibuca OR cortes OR colon OR ocotepeque OR "Gracias a Dios" OR atlantida OR "Islas de la Bahia" OR nicaragua OR nicoya OR boaco OR carazo OR chinandega OR chontales OR esteli OR granada OR jinotega OR leon OR madriz OR managua OR masaya OR matagalpa OR "Nueva Segovia" OR rivas OR "Rio San Juan" OR "Costa Caribe Norte" OR "North Caribbean Coast" OR "Costa Caribe Sur" OR "South Caribbean Coast" OR "Costa Rica" OR alajuela OR cartago OR guanacaste OR heredia OR limon OR puntarenas OR "San Jose" OR panama OR veraguas OR chiriqui OR "Los Santos" OR cocle OR herrera OR "Bocas del Toro" OR darien OR "Kuna Yala" OR "Panama Oeste" OR "Panama West") AND (Lake OR lago OR bog OR pantano OR swamp OR pantano OR cave OR cueva OR estuary OR estuario OR dendro OR terrestrial OR terrestre OR marine OR marina OR "Closed basin" OR "Cuenca Cerrada" OR "Open basin" OR "Cuenca abierta" OR fluvial OR aeolian OR eolico OR speleothem OR core OR nucleo OR sediment OR sedimento OR midden OR muladar OR limnology OR limnologia OR climatology OR climatologia OR dendrochronology OR dendrocronologia OR dendroclimatology OR dendroclimatologia OR palaeo OR paleo OR climatology OR climatologia OR ecological OR ecologico OR ecology OR ecologia OR environment OR ambiente OR palynological OR palynology OR palaeoecology OR paleoecology) AND (Holocene OR holoceno OR "Little Ice Age" OR "Pequena Edad de Hielo" OR lia OR "Medieval Climate Anomaly" OR "Anomalia del clima medieval" OR mca OR "Medieval Warm Period" OR "Periodo calido medieval" OR mwp OR "Palaeo Indian" OR archaic OR arcaico OR "Early Preclassic" OR "Preclasico Temprano" OR "Middle Preclassic" OR "Preclasico Medio" OR "Late Preclassic" OR "Preclasico Tardio" OR "Early Classic" OR "Clasico Temprano" OR "Late Classic" OR "Clasico Tardio" OR "Terminal Classic" OR "Clasico Terminal" OR "Early Post Classic" OR "Posclasico Temprano" OR "Late Post Classic" OR "Posclasico Tardio" OR "Colonial Period" OR "Spanish Conquest" OR "Conquista Espanola" OR "Periodo de contacto" OR "Independent Mexico" OR "Mexico Independiente" OR "Time series" OR "Series de tiempo" OR temporal OR chronology OR cronologia OR chron OR cron OR anthropocene OR antropoceno OR history OR historia OR prehistory OR prehistoria OR quaternary OR cuaternario) AND (Pollen OR polen OR "Fossil pollen" OR "Polen fosil" OR "Macro charcoal" OR "Macro carbon" OR "Micro charcoal" OR "Micro carbon" OR isotope OR isotopo OR do18 OR "Tree ring" OR "Anillo de arbol" OR chironomid OR diatom OR diatomea OR ostracod OR xrf OR "X-ray fluorescence" OR "Fluorescencia de rayos X" OR geochemistry OR geoquimica OR "Loss on ignition" OR "Perdida por ignicion" OR loi OR coleoptera OR coleoptero OR "Magnetic Susceptibility" OR "Susceptibilidad magnetica" OR "Sporormiella" OR "dung fungal spores" OR "Estiercol de hongos estiercol" OR macrofossil OR macrofosil OR vegetation OR vegetacion OR burning OR fuego OR plant OR planta OR tree OR arbole OR shrub OR arbusto OR herb OR hierba OR recovery OR recuperacion OR resilience OR resistencia OR disturbance OR disturbio OR reconstruction OR reconstruccion OR "Land use" OR "Uso del tierra" OR human OR humano OR civilization OR civilizacion

## Appendix B. Test Library

Anderson, L. and Wahl, D. Two Holocene paleofire records from Peten, Guatemala: Implications for natural fire regime and prehispanic Maya land use. *Glob. and Planetary Change* **2016**, *138*, 82–92.

Avnery, S., Dull, R.A. and Keitt, T.H. Human versus climatic influences on late-Holocene fire regimes in southwestern Nicaragua. *Holocene*, **2011**, *21*(4), 699–706.

Bush, M.B. and Colinvaux, P.A. Tropical forest disturbance: paleoecological records from Darien, Panama. *Ecology* **1994**, *75*(6), 1761–1768.

Clement, R.M. and Horn, S.P. Pre-Columbian land-use history in Costa Rica: a 3000-year record of forest clearance, agriculture and fires from Laguna Zoncho. *Holocene* **2001**, *11*(4), 419–426.

Correa-Metrio, A., Vélez, M.I., Escobar, J., St-Jacques, J.M., López-Pérez, M., Curtis, J. and Cosford, J. Mid-elevation ecosystems of Panama: future uncertainties in light of past global climatic variability. *J. Quat. Sci.* **2016**, *31*(7), 731–740.

Dull, R.A. A Holocene record of Neotropical savanna dynamics from El Salvador. *J. Paleolimnol.* **2004**, *32*(3), 219–231.

Dull, R.A., An 8000-year record of vegetation, climate, and human disturbance from the Sierra de Apaneca, El Salvador. *Quat. Res.* **2004**, *61*(2), 159–167.

Hillesheim, M.B., Hodell, D.A., Leyden, B.W., Brenner, M., Curtis, J.H., Anselmetti, F.S., Ariztegui, D., Buck, D.G., Guilderson, T.P., Rosenmeier, M.F. and Schnurrenberger, D.W. Climate change in lowland Central America during the late deglacial and early Holocene. *J. Quat. Sci.* **2005**, *20*(4), 363–376.

Horn, S.P., Rodgers III, J.C., Orvis, K.H. and Northrop, L.A. Recent land use and vegetation history from soil pollen analysis: testing the potential in the lowland humid tropics. *Palynology* **1998**, *22*(1), 167–180.

Islebe, G.A. and Hooghiemstra, H. Vegetation and climate history of montane Costa Rica since the last glacial. *Quat. Sci. Rev.* **1997**, *16*(6), 589–604.

Lachniet, M.S., Burns, S.J., Piperno, D.R., Asmerom, Y., Polyak, V.J., Moy, C.M. and Christenson, K., A 1500-year El Niño/Southern Oscillation and rainfall history for the isthmus of Panama from speleothem calcite. *J. Geophys. Res.* **2004**, *109*, D20117.

Metcalfe, S., Breen, A., Murray, M., Furley, P., Fallick, A. and McKenzie, A. Environmental change in northern Belize since the latest Pleistocene. *J. Quat. Sci.* **2009**, *24*(6), 627–641.

Neff, H., Pearsall, D.M., Jones, J.G., Arroyo, B., Collins, S.K. and Freidel, D.E. Early Maya adaptive patterns: Mid-late Holocene paleoenvironmental evidence from Pacific Guatemala. *Lat. Am. Antiq.* **2006**, *17*(3), 287–315.

Piperno, D.R., Bush, M.B. and Colinvaux, P.A. Paleoenvironments and human occupation in late-glacial Panama. *Quat. Res.* **1990**, *33*(1), 108–116.

Piperno, D.R., Bush, M.B. and Colinvuax, P.A. Paleoecological perspectives on human adaptation in Central Panama. II The Holocene. *Geoarchaeology* **1991**, *6*(3), 227–250.

Pohl, M.D., Pope, K.O., Jones, J.G., Jacob, J.S., Piperno, D.R., Lentz, D.L., Gifford, J.A., Danforth, M.E. and Josserand, J.K. Early agriculture in the Maya lowlands. *Lat. Am. Antiq.* **1996**, *7*(4), 355–372.

Slate, J.E., Johnson, T.C. and Moore, T.C. Impact of pre-Columbian agriculture, climate change, and tectonic activity inferred from a 5,700-year paleolimnological record from Lake Nicaragua. *J. Paleolimnol.* **2013**, *50*(1), 139–149.

Taylor, Z.P., Horn, S.P. and Finkelstein, D.B. Maize pollen concentrations in Neotropical lake sediments as an indicator of the scale of prehistoric agriculture. *Holocene* **2013**, *23*(1), 78–84.

Urquhart, G.R. Paleoecological record of hurricane disturbance and forest regeneration in Nicaragua. *Quat. Int.* **2009**, *195*(1–2), 88–97.

Wahl, D., Hansen, R.D., Byrne, R., Anderson, L. and Schreiner, T. Holocene climate variability and anthropogenic impacts from Lago Paixban, a perennial wetland in Peten, Guatemala. *Glob. Planet. Chang.* **2016**, *138*, 70–81.

Whitmore, T.J., Brenner, M., Curtis, J.H., Dahlin, B.H. and Leyden, B.W. Holocene climatic and human influences on lakes of the Yucatan Peninsula, Mexico: an interdisciplinary, palaeolimnological approach. *Holocene*, **1996**, *6*(3), 273–287.

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
