# Peer review of "Forests, Water, and Land Use Change across the Central American Isthmus: Mapping the Evidence Base for Terrestrial Holocene Palaeoenvironmental Proxies"

_forests, doi:10.3390/f12081057_

Round 1
Reviewer 1 Report
Insightful and robust evaluation of evidence base for anthropogenic use of forests across Central America during the Holocene period.
Systematic evidence evaluation is a proven suitable method of assessing bundles of scientific data and findings, and this contribution is a most welcome and comprehensive addition to the regional field.
395-96: The literature identification mentions only 0.7% of articles published in Spanish. Could this be a selection bias? Surely, there should be more available in regional scholarship periodicals. In how far is language difference the issue in the selection process?
403-04 check capitalising of mentioned nouns
444 What is happening at the right end of the graph? Lines appear to drop precipitously? Is that a technical representation issue due to the 20-year time slices?
504 Figure is insufficiently legible. Location can be made out (approx.) but number of studies is impressionistic.
511 Sentence full-stop is missing.
520 Consistency of decimal use would stipulate '0.0%', not '0%' and so on for all integer numbers.
534 Excess space preceding 'the'? See also 535.
566 Preclassic is understood to run up to early centuries CE, say 250 CE.
590 'Maya empire' is antiquated. Preferably refer to Classic period Maya city-states
630 Single quotation missing. The mentioned journals are indeed likely to yield additional studies. Overall, a clear listing of which journals were included would improve clarity of the remit of the evidence base used for this study.
631 Hard return issue?
General typographical comment: Should time range be indicated by an en-dash rather than a hyphen?
Author Response
The authors would like to thank Reviewer 1 for their helpful comments and careful attention to detail. All suggestions from this reviewer have been incorporated into the main text.
- 395-96: The literature identification mentions only 0.7% of articles published in Spanish. Could this be a selection bias? Surely, there should be more available in regional scholarship periodicals. In how far is language difference the issue in the selection process?
There is a notable dearth of Spanish language publications and while this literature is likely to exist in regional institutions, this scholarship is not recorded in the databases queried for this systematic map. This is a common limitation in all reviews and is always noted as such in systematic reviews. The authors agree that this is a limitation and have acknowledged this limitation in L621-637.
- 403-04 check capitalising of mentioned nouns
Capitalisation of mentioned nouns corrected.
- 444 What is happening at the right end of the graph? Lines appear to drop precipitously? Is that a technical representation issue due to the 20-year time slices?
Many authors report their data starting from 0 B.P. (Before Present) which translates to the year 1950. This is a widely used convention for radiometrically dated sequences. The presented data is correct.
- 504 Figure is insufficiently legible. Location can be made out (approx.) but number of studies is impressionistic.
We have replaced figure 10 with a larger more legible figure.
- 511 Sentence full-stop is missing.
Full-stop added.
- 520 Consistency of decimal use would stipulate '0.0%', not '0%' and so on for all integer numbers.
Integer consistency corrected.
- 534 Excess space preceding 'the'? See also 535.
Excess spaces removed
- 566 Preclassic is understood to run up to early centuries CE, say 250 CE.
Corrected to 250C.E.
- 590 'Maya empire' is antiquated. Preferably refer to Classic period Maya city-states
Corrected in text.
- 630 Single quotation missing. The mentioned journals are indeed likely to yield additional studies. Overall, a clear listing of which journals were included would improve clarity of the remit of the evidence base used for this study.
The authors agree with the reviewer that the mentioned missing journals are likely to contain additional information; however, in keeping with the methodological protocol the authors did not include grey literature or dark data. This is stated in the limitations section L621-637. For a list of all journals picked up from the three online bibliographic databases please refer to the supplementary evidence base spreadsheet column F.
- 631 Hard return issue?
Thank you, corrected.
- General typographical comment: Should time range be indicated by an en-dash rather than a hyphen?
The authors have changed all hyphens relating to time ranges to en dash.
Reviewer 2 Report
Harvey et al. review (Forests, Water, and Land Use Change across the Central American Isthmus: Mapping the evidence base for terrestrial Holocene palaeoenvironmental proxies)
I found this paper very confusing. After reading the Abstract, I was unsure about what the goals of the study were or what was ultimately accomplished. Similarly, I did not understand the title. But this may simply reflect my lack of familiarity with terms like “evidence base.” Throughout the manuscript, I felt that wording was often imprecise and therefore confusing. I will point out selected cases, which I think create confusion. I will proceed by line number, which I hope will be helpful for the authors.
Line
15 – “Climatic and anthropogenic drivers currently provide the largest threats to changes in the forest cover and composition for this region” Comment: clearly, the drivers are not “threats to changes,” rather, what I think you want to say is “Climate and humans are the main drivers of change in forest cover and composition in this region” or “Climatic and anthropogenic drivers threaten to cause changes”
17 – “therefore understanding the dynamics of these systems, their variability across space and through time is important for discerning current and future responses.” It is a little unclear what the word “responses” refers to here, but I presume you mean “predicting how forest cover and composition will respond to future climate change and human activities.”
21 – data “were identified and mapped following best practice for systematic evidence synthesis.” Not sure what that means.
22 – “Results from the evidence base were summarised to show the spatial and temporal extent of contemporary research.” Again, I am not sure what is meant by an “evidence base.” In the second part of the sentence, what do you mean by “contemporary research?” Or are you really referring to “existing data from previous studies”?
25 – What is meant by “relevance at full-text”?
34 – “and significantly affect the ability to draw unbiased conclusions” – about what?
35-41 – These are the major conclusions of the study, correct? If I understand, the problem with many of the previous studies is that data reporting was not standardized, transparent or repeatable. But then there is a call for obtaining raw data from previous studies, which can be used to target specific questions, but I don’t think “standardizing reporting” should be included, as that is mentioned previously. Are these being targeted to better understand the paleo record? Why do you refer to the “the role and impacts of fire and burning”? Why not just “the impacts of fire”? Or did you intend to distinguish between natural and human-caused fires? Is it necessary to follow “changes in hydroclimate” with “water availability, and risk of tropical storms”? Aren’t they part of hydroclimate? Also, why not add risk of droughts?
52 – “and their impacts to forests” the previous part of the sentence refers to the “dynamics of these systems” so what does “their refer to?”
63 – “primera”?
76-82 – These lines seem to be key and maybe belong, in some fashion, in the Abstract.
84 – Why “distribution of the forests”? Why not simply “Forests across the Central America Isthmus can be broadly assigned to six terrestrial biomes:”?
90 – I think you should define, specifically, how you distinguish moist from dry forests (give annual rainfall values?). I have often seen Peten (Guatemala) forests referred to as dry tropical forests (there is some deciduousness), but you relegate them to the moist forest biome.
104 – “anthropogenically desirable” is odd. Reorganize the sentence to be more straightforward, e.g. “This region is densely populated because it is ideal for shifting agriculture and cattle ranching.”
113 – What is meant by “complex successional interactions”?
118 – Extensive areas of Peten, Guatemala, also support savannas
122 – I suspect you mean “Motagua”
128 – The common name “black mangrove” usually refers to Avicennia germinans. I do not know of a mangrove called Terminalia erectus, but maybe you are thinking of Conocarpus erectus (button mangrove). It tends to grow farther inland, sometimes around lakes, and is relatively shorter in stature. The genusTerminalia (Combretaceae) refers to what is locally called almendra (almond).
143 – “greatest variability in precipitation between seasons”? Do you mean “greatest difference in precipitation between seasons”?
143 – I confess, I have no idea what is meant by “Covariance between seasonal precipitation.” Could you briefly define that? The units on the plot (Low:1000 to High:10000) provide no hint. Note that “Covariance” is spelled wrong on the plot.
150 – change “between” to “among” and hyphenate “Climate-driven”
156 – what is meant by “increased magnitudes of burning”? Do you mean more fires or more intense fires?
158-160 – With respect to how many times forests have burned, what time frames do you refer to here?
162 – are you using “c.” here, before “50 kW” to mean about? Use of “c.” or “ca.” should be reserved for time. You can use “~”. Also, change the area to “m-2”
172 – superscript “-2”
173 – Do you mean “intense”
178 – delete “different”
180 – “and controlling ignitions igniting more or fewer fires irrespective of season or weather”. Maybe change to “and controlling ignitions, lighting more or fewer fires regardless of season or weather”.
182 – “anthropogenically initiated” – why not “the Indio Maiz forest fire, apparently started by local people, destroyed…”
185 – “anthropogenic settlement” – what other type is there?
196 – change to “From 1966 to 1994”
197 – change to “between 2001 and 2010”
197 – This may deserve another sentence. This implies that Guatemala lost 22% of its forest cover in the interval 1966-1994 and another 70% from 2001 to 2010 (7%/yr). I assume some was lost in the interval 1994-2000. So there is virtually none left, correct?
198 – “Across northern Guatemala this deforestation is strongly associated with distance to human settlements and rapid population growth.” I imagine that the deforestation is negatively associated with distance to settlements and positively related with population?
202-203 – I think you have to make the contrast between Guatemala/Nicaragua and Honduras/El Salvador much stronger. For instance, “In contrast to the forest loss in Guatemala and Nicaragua during the decade 2001-2010, Honduras and El Salvador experienced forest recovery, as 2335 km2 of agricultural land was abandoned.”
208 – “however, there are a variety of challenges associated with interpreting these data sets enabling human impacts to be disentangled from natural drivers.” How about ““however, it can be challenging to use such data sets to disentangle human from natural drivers of forest change.”
213 – “maize”
220 – “Determining if a given taxa is in situ or anthropogenically introduced and cultivated is often a topic of debate within papers and is discussed within the context of the most likely scenario”? How about “It can be difficult to determine if a taxon was native and grew naturally or was introduced and cultivated, and authors often simply report the most likely scenario”
226-228 – Italicize “Zea mays” and “Tripsacum”
229 – “it is used to suggest anthropogenic presence” – How about ”it suggests past human presence.”
230 – “In order to distinguish between anthropogenic and climate driven burning it is important to take a multi-proxy approach by using independent proxies to reconstruct climate, vegetation, and fire and where possible use multiple proxies to re-construct fire.” I am not a big fan of “reconstruct” (I prefer “infer”), but beyond that, I would break that into two sentences and be clearer.
241 – “past climatic reviews” – do you mean “review papers about past climate”?
243 – “onset into the Holocene” – why not “Early Holocene”?
243 – here and elsewhere in the paragraph add a space between the date and B.C.E. or C.E.
257 – “To date, there has been no attempt to systematically map the evidence base related to palaeoecological proxies in the region.” Maybe I am just not familiar with the approach, but I do not know what that means. And why “related to”?
258 – “The current research was therefore undertaken to provide a systematic evidence evaluation of relevant proxies and to organise the evidence base assembled in this process in an accessible way for future land-use policy- and decision-making.” Not sure what any of that means.
273 – “While systematic reviews and systematic maps have mostly been adopted for policy-relevant questions, the methods that dictate their conduct are of interest to academic literature reviews generally, given that transparency and repeatability are the corner-stones of science.” I am not sure what a systematic map is, what its “conduct” is.
Table 1 – What are PICO and PECO? OK, I get it – Population, Exposure, Counterfactual, Outcomes
298 – I am not sure what a “search string” is.
303 – note that “palaeoenvironmental” is spelled wrong. Also note that it is spelled differently throughout the article (w or w/o “a”)
304 – So what is the rationale for using Boolean operators? Why did you do that?
305 – could you simply say “Geographic location was constrained to the region between the Isthmus of Tehuantepec and the Isthmus of Panama.”?
306 – change to “10 August 2020”
314 – Not sure what is meant by “Search strings were refined several times to remove terms containing generic nomenclature while still maintaining all relevant papers.”
318 – Also confused by “Search terms were iteratively modified and optimised until the entire Test Library was returned.”
Table 4 – “Environmental proxy data must be representative of a point in time.” and “Environmental proxy data must be representative of different a point in time.” Maybe mention why these are criteria for inclusion.
327-336 – I was confused by the mix of past and future tense.
355 – “cross cutting lens”?
Table 5 and elsewhere – keep in mind that “data are”
359 – “Metadata documenting the reproducibility of the study were extracted to give an indication of the quality of the research design; however, in common with most systematic evidence maps, critical appraisal of all studies selected at full-text was not undertaken and no studies were excluded from the evidence base on the basis of likelihood of bias.” I am unclear about how metadata document the reproducibility of a study.
Fig. 4 Note that the number of “Duplicates removed” in the figure (3508) is different from the value on line 366 of the text (3502).
Section 3.3 – I would get rid of all the “#” and either use nothing or “n=XXX”
Figure 6 legend – “Number of studies extracted from different environmental archives.” I don’t think studies are extracted from archives.
429 – “The number of palaeoenvironmental records available increases throughout the Holocene.” I don’t think this is what you mean to say (clearly most of the studies only became available in the last few decades). So reword for clarity.
434 – “These studies are considered low risk in terms of good methodological practice for sequence dating.” I am not familiar with this jargon, but what is meant by “risk” in this context?
438 – “High risk studies (#205, 36.4%) are studies which have not dated the bottom of their sequence (Figure 9A). This is particularly problematic for studies which have been presented in their publications as measurements of depth only (#317; 58.9%) when they are discussed as time (Figure 9B), and this is further complicated when the raw study data is not available with the published article (#496, 92.2%) (Figure 9C).”
Change “which” to “that” in several places (lines 438-439). I am not sure “studies” date sequences. What does “This” refer to in the second sentence? Also “studies which have been presented in their publications as measurements of depth only” – do you mean “for studies in which published data are reported only with respect to depth in the sequence, but are discussed in terms of time?” Not really certain what was intended.
Figure 8. I think the legend need some clarification. Why is “proxies” in parentheses after “studies”? You should label the y axis. I gather that “forest, land use change, and water” are what was inferred in these studies.
Figure 9. “Percentage of studies with a high, medium or low risk for sequence dating.” Again, I am unsure of what is meant by “risk.”
Table 6. I don’t think the header says what you want it to say. Maybe something like “Analyzed variable in paleoenvironmental archives and the environmental variable(s) for which it served as a proxy.” Also, I think there are only 17 such proxy variables, not 18, as Testate amoebae and Thecamoebians are one and the same.
483 – Change “Relative” to “Relatively”
488 – Change “while” to “whereas”
499 – what is meant by “soil sediments”?
510 – “There has been a particular increase in the number of studies published surrounding the last decade (49.5%).” What does that mean? Literally, it seems to me that these studies deal with changes between 2011 and today.
511 – Need a period before “Spatially”
515 – “Evidence for forests are”?
543 – “however, the absence of this data with the published article presents a worrying barrier to future access, use and quality assessment of these data sets.”?
549 – “The guidelines and quality of dating practices for palaeoenvironmental research across the Central American Isthmus is recommended for review.” Not sure what you propose here. Are there “guidelines”?
550 – “Accurately dating and presenting palaeoenvironmental studies, inclusive of the events represented within them, presents significant uncertainty.” So, if we ignore the middle clause, you are saying that “Accurately dating studies presents significant uncertainty.” How can that be? I would think that accurate dating would eliminate uncertainty. But furthermore, I do not think one “dates studies.”
552 – “This is due to a number of factors including good methodological practice but also:” So I read that to mean that the “significant uncertainty” is caused by “good methodological practice.”
556 – do you mean “variance” or “differences”?
568 – “Identifying the impacts of different land uses across space and time would contribute towards our understanding of the long-term impacts to forest community composition and arboreal cover.” Yes, but somewhere I think it is critical to mention that the causes for shift in forest cover/composition today are very different from what they were in the past. Today the forest is removed with chain saws and much of the cleared land is used for cattle ranching. Pre-Columbian people worked with stone tools and fire, and there were no large domestic herbivores.
585 – you do not need the hyphen with words ending in “ly”
606 – “to our understanding of different types of ecosystem resilience”? What are different types of ecosystem resilience?
607 – “Using the palaeoecological records identified in this study, disturbance events and rates of recovery could be quantitatively identified and analysed from high resolution data sets using locally defined thresholds.” Not sure what you mean by “locally defined thresholds”? How will those be established?
614-619 – what does “key topics” refer to? “to inform on land-use policy”? I think you mean “to inform land-use policy”
“Integrating modern measurements, historical records, and palaeoenvironmnental data would enable quantifiable estimates of the above impacts and provide the knowledge base to make informed policy decisions in the future.” What does “above impacts” refer to? Aren’t “(a) climate change, (b) increasing populations pressures, and (c) land use change” the drivers of what you are interested in measuring in the paleo archives?
I hope that another reviewer, who is more familiar with the approaches used in this article, can weigh in. I think the piece could benefit from a complete overhaul, in which the objectives are clearly stated and the findings, re those objectives, are clearly presented. It is the prerogative of the authors to decide what their goals were in undertaking this study, but I think they muddy the waters by including boiler plate verbiage about how such paleo information can be used for risk mitigation, planning and conservation. Maybe I am just cynical, but I am dubious that the data can (or will be) used in such a fashion. The socio-political situation today in much of Central America will simply prevent the success of effective conservation efforts in many cases. I am a “believer” in Conservation Paleobiology, but its utility can be overestimated. In any case, is it not sufficient to say that you undertook this study to evaluate the information that is out there regarding forest dynamics in CA throughout the Holocene? Seems to me that the history of such tropical biotic changes is inherently interesting. And why don’t you get rid of “water” and “land use change” in the title? Aren’t those the drivers of changes in forest cover and composition? Maybe I am wrong, but it seems to me that the greatest rationale for this analysis of the existing literature was to identify what further work is required to build a clearer picture of Holocene vegetation change in Central America, and make suggestions about how that can be accomplished. Apparently, there are data gaps with respect to time intervals, geographic areas and forest types studied. But as I understand, one of the big problems is that raw data for most studies do not accompany the original publications. Do you suspect that access to that raw data would alleviate the need for more studies in neglected areas or time intervals? Anyway, I think considerable effort went into this analysis and I think it would be helpful if the findings were presented in a simple, straightforward way.
Author Response
The authors would like to thank Reviewer 2 for their comments which have led to the improvement of this manuscript. The majority of the Reviewer’s comments and suggestions have been included in the manuscript with clarification provided where we disagree with the Reviewer.
- 15 – “Climatic and anthropogenic drivers currently provide the largest threats to changes in the forest cover and composition for this region” Comment: clearly, the drivers are not “threats to changes,” rather, what I think you want to say is “Climate and humans are the main drivers of change in forest cover and composition in this region” or “Climatic and anthropogenic drivers threaten to cause changes”
We have changed L15 following the reviewer’s recommendation: “Climatic and anthropogenic drivers threaten to cause changes in the forest cover…”
- 17 – “therefore understanding the dynamics of these systems, their variability across space and through time is important for discerning current and future responses.” It is a little unclear what the word “responses” refers to here, but I presume you mean “predicting how forest cover and composition will respond to future climate change and human activities.”
Correct.
- 21 – data “were identified and mapped following best practice for systematic evidence synthesis.” Not sure what that means.
Systematic Mapping is an established method used to collect, collate, and present research evidence. More information can be found here: https://environmentalevidence.org/information-for-authors/. The Special Issue is dedicated to papers adopting or adapting systematic review methods (see https://www.mdpi.com/journal/forests/special_issues/systematic_methods) and the Editors have welcomed submissions that deal with established uses of the best practice guidelines as well as novel papers presenting approaches that use the methodologies in fields that have not used these techniques to date.
- 22 – “Results from the evidence base were summarised to show the spatial and temporal extent of contemporary research.” Again, I am not sure what is meant by an “evidence base.” In the second part of the sentence, what do you mean by “contemporary research?” Or are you really referring to “existing data from previous studies”?
An evidence base refers to the collection of articles returned by following the stated protocol. We have clarified the sentence following the reviewer’s suggestion: “Results from the evidence base were summarised to show the spatial and temporal extent of the published data sets.”
- 25 – What is meant by “relevance at full-text”?
Relevance at full text refers to the screening process undertaken when following a systematic evidence evaluation. Please see: https://environmentalevidence.org/information-for-authors/6-eligibility-screening/. On advice from the Editors during the process of considering our submissions, they argued against each individual paper including a full description of the elements of a systematic review. This will be covered by the Special Issue editorial.
- 34 – “and significantly affect the ability to draw unbiased conclusions” – about what?
We have removed the word unbiased to help clarify the sentence.
- 35-41 –
- These are the major conclusions of the study, correct?
Correct.
- If I understand, the problem with many of the previous studies is that data reporting was not standardized, transparent or repeatable. But then there is a call for obtaining raw data from previous studies, which can be used to target specific questions, but I don’t think “standardizing reporting” should be included, as that is mentioned previously. Are these being targeted to better understand the paleo record?
The authors strongly believe in their recommendation of standardised reporting for all palaeoenvironmental data. We believe that it will open a discussion, that may not agree with our view, but that the discussion itself on standardisation – the practicalities or indeed desirability- will be useful in the field. The usefulness of palaeo data can be increased across multiple fields including, but not limited to, ecology, forestry, archaeology, and policy - if the data are comparable. The authors acknowledge the challenges of standardising past data sets; however, if these data sets are incomplete by modern standards, then (as a discipline/field) we should consider re-collecting and analysing data from these sites.
- Why do you refer to the “the role and impacts of fire and burning”? Why not just “the impacts of fire”? Or did you intend to distinguish between natural and human-caused fires?
The authors refer to the impacts of fire and burning to capture the role of natural and human-caused fire; however, the authors do not attempt to distinguish between the two. Please see L230-233.
- Is it necessary to follow “changes in hydroclimate” with “water availability, and risk of tropical storms”? Aren’t they part of hydroclimate? Also, why not add risk of droughts?
The authors acknowledge the overlap between “hydroclimate” and the other terms; however, believe these should remain separate to cover all aspects of “water”, including seawater and individual events such as tropical storms.
- 52 – “and their impacts to forests” the previous part of the sentence refers to the “dynamics of these systems” so what does “their refer to?”
This sentence has been clarified to: “Understanding the dynamics of these systems, their variability across space and through time, and the impacts of climate and anthropogenic drivers to forests, is therefore important for discerning current and future responses.”
- 63 – “primera”?
“Pimera” has been clarified in the main body of the text (L65) as the start of the agricultural season.
- 76-82 – These lines seem to be key and maybe belong, in some fashion, in the Abstract.
These lines have been incorporated into the Abstract.
- 84 – Why “distribution of the forests”? Why not simply “Forests across the Central America Isthmus can be broadly assigned to six terrestrial biomes:”?
We have updated line 84 with the reviewer’s suggestion: “Forests across the Central America Isthmus…”
- 90 – I think you should define, specifically, how you distinguish moist from dry forests (give annual rainfall values?). I have often seen Peten (Guatemala) forests referred to as dry tropical forests (there is some deciduousness), but you relegate them to the moist forest biome.
The forest types across Central America have been defined according to the biomes reported in Dinerstein et al. (2017). Please see Figure 1 (L94).
- 104 – “anthropogenically desirable” is odd. Reorganize the sentence to be more straightforward, e.g. “This region is densely populated because it is ideal for shifting agriculture and cattle ranching.”
The authors have updated this sentence with the reviewer’s suggestion.
- 113 – What is meant by “complex successional interactions”?
Complex successional interactions refer to the compositional transition of forest species and how they impact each other. We use the standard definition of ‘succession’ as defined by John Helms in ‘The Dictionary of Forestry’ (1998, Society of American Foresters, publ. CABI) – “the gradual supplanting of one community of plants by another”, and his definitions of allogenic and autogenic succession to imply, as other authors have that these are complex processes in the region (e.g., Corrales, Bouroncle & Zamora (2015) who build on Kappelle 2006 and Rzedowski 2006, whom we reference.
- 118 – Extensive areas of Peten, Guatemala, also support savannas
The authors acknowledge the reviewer’s point and would draw their attention to the authors’ source of data (Dinerstein et al. 2017) for defining these biomes.
- 122 – I suspect you mean “Motagua”
Thank you, please see recommended correction in the manuscript.
- 128 – The common name “black mangrove” usually refers to Avicennia germinans. I do not know of a mangrove called Terminalia erectus, but maybe you are thinking of Conocarpus erectus(button mangrove). It tends to grow farther inland, sometimes around lakes, and is relatively shorter in stature. The genusTerminalia (Combretaceae) refers to what is locally called almendra (almond).
The authors would like to thank the reviewer for bringing this taxonomic inconsistency to their attention. All taxonomic synonyms have been updated following the Plant List: http://www.theplantlist.org/
- 143 – “greatest variability in precipitation between seasons”? Do you mean “greatest difference in precipitation between seasons”?
The authors have updated the manuscript with the reviewer’s suggestion.
- 143 – I confess, I have no idea what is meant by “Covariance between seasonal precipitation.” Could you briefly define that? The units on the plot (Low:1000 to High:10000) provide no hint. Note that “Covariance” is spelled wrong on the plot.
The authors have updated the spelling mistake in Figure 2 and have added a brief description defining covariance between seasonal precipitation – please see caption of Figure 2.
- 150 – change “between” to “among” and hyphenate “Climate-driven”
The authors have updated the manuscript with the reviewer’s suggestions.
- 156 – what is meant by “increased magnitudes of burning”? Do you mean more fires or more intense fires?
The authors have removed the word “magnitudes” to avoid any confusion and have updated this sentence.
- 158-160 – With respect to how many times forests have burned, what time frames do you refer to here?
The authors have clarified this sentence to make clear times burnt within a successional forest cycle (mainly fires of 1995 and 1996). Cochrane’s earlier paper provides additional information about the region, which was not included in the systematic map because it did not meet inclusion criteria. This reference is in the Background section, and as such, we are setting the scene rather than interpreting Cochrane’s points.
- 162 – are you using “c.” here, before “50 kW” to mean about? Use of “c.” or “ca.” should be reserved for time. You can use “~”. Also, change the area to “m-2”
The authors have updated the manuscript with the reviewer’s suggestions.
- 172 – superscript “-2”
The authors have updated the manuscript with the reviewer’s suggestions.
- 173 – Do you mean “intense”
The authors stand by the use of the word intensity.
- 178 – delete “different”
The authors have updated the manuscript with the reviewer’s suggestions.
- 180 – “and controlling ignitions igniting more or fewer fires irrespective of season or weather”. Maybe change to “and controlling ignitions, lighting more or fewer fires regardless of season or weather”.
The authors have updated the manuscript with the reviewer’s suggestions.
- 182 – “anthropogenically initiated” – why not “the Indio Maiz forest fire, apparently started by local people, destroyed…”
The authors have updated the manuscript with the reviewer’s suggestions.
- 185 – “anthropogenic settlement” – what other type is there?
Good point. The authors have removed the word anthropogenic.
- 196 – change to “From 1966 to 1994”
The authors have updated the manuscript with the reviewer’s suggestions.
- 197 – change to “between 2001 and 2010”
The authors have updated the manuscript with the reviewer’s suggestions.
- 197 – This may deserve another sentence. This implies that Guatemala lost 22% of its forest cover in the interval 1966-1994 and another 70% from 2001 to 2010 (7%/yr). I assume some was lost in the interval 1994-2000. So there is virtually none left, correct?
The authors have clarified this sentence in the manuscript.
- 198 – “Across northern Guatemala this deforestation is strongly associated with distance to human settlements and rapid population growth.” I imagine that the deforestation is negatively associated with distance to settlements and positively related with population?
Correct.
- 202-203 – I think you have to make the contrast between Guatemala/Nicaragua and Honduras/El Salvador much stronger. For instance, “In contrast to the forest loss in Guatemala and Nicaragua during the decade 2001-2010, Honduras and El Salvador experienced forest recovery, as 2335 km2of agricultural land was abandoned.”
The authors have updated the manuscript with the reviewer’s suggestion.
- 208 – “however, there are a variety of challenges associated with interpreting these data sets enabling human impacts to be disentangled from natural drivers.” How about ““however, it can be challenging to use such data sets to disentangle human from natural drivers of forest change.”
The authors have updated the manuscript with the reviewer’s suggestion.
- 213 – “maize”
The authors have corrected the capitalisation.
- 220 – “Determining if a given taxa is in situ or anthropogenically introduced and cultivated is often a topic of debate within papers and is discussed within the context of the most likely scenario”? How about “It can be difficult to determine if a taxon was native and grew naturally or was introduced and cultivated, and authors often simply report the most likely scenario”
The authors have updated the manuscript with the reviewer’s suggestion.
- 226-228 – Italicize “Zea mays” and “Tripsacum”
The authors have updated the manuscript with the reviewer’s suggestion.
- 229 – “it is used to suggest anthropogenic presence” – How about ”it suggests past human presence.”
The authors have updated the manuscript with the reviewer’s suggestion.
- 230 – “In order to distinguish between anthropogenic and climate driven burning it is important to take a multi-proxy approach by using independent proxies to reconstruct climate, vegetation, and fire and where possible use multiple proxies to re-construct fire.” I am not a big fan of “reconstruct” (I prefer “infer”), but beyond that, I would break that into two sentences and be clearer.
The authors have updated the manuscript with the reviewer’s suggestions.
- 241 – “past climatic reviews” – do you mean “review papers about past climate”?
The authors have updated the manuscript with the reviewer’s suggestions.
- 243 – “onset into the Holocene” – why not “Early Holocene”?
The authors have updated the manuscript with the reviewer’s suggestions.
- 243 – here and elsewhere in the paragraph add a space between the date and B.C.E. or C.E.
The authors have updated the manuscript with the reviewer’s suggestions.
- 257 – “To date, there has been no attempt to systematically map the evidence base related to palaeoecological proxies in the region.” Maybe I am just not familiar with the approach, but I do not know what that means. And why “related to”?
The authors have clarified this sentence in the manuscript.
- 258 – “The current research was therefore undertaken to provide a systematic evidence evaluation of relevant proxies and to organise the evidence base assembled in this process in an accessible way for future land-use policy- and decision-making.” Not sure what any of that means.
The authors have clarified this sentence in the manuscript.
- 273 – “While systematic reviews and systematic maps have mostly been adopted for policy-relevant questions, the methods that dictate their conduct are of interest to academic literature reviews generally, given that transparency and repeatability are the corner-stones of science.” I am not sure what a systematic map is, what its “conduct” is.
Systematic Mapping is an established method used to collect, collate, and present research evidence. More information can be found here: https://environmentalevidence.org/information-for-authors/
- Table 1 – What are PICO and PECO? OK, I get it – Population, Exposure, Counterfactual, Outcomes
PICO is another standard systematic review framework: Population, Intervention, Counterfactual, Outcomes.
- 298 – I am not sure what a “search string” is.
These are the searched key words jointed together with Boolean operators. The authors have added clarification to the manuscript.
- 303 – note that “palaeoenvironmental” is spelled wrong. Also note that it is spelled differently throughout the article (w or w/o “a”)
The authors have standardised the spelling of “palaeoenvironmental” following British convention.
- 304 – So what is the rationale for using Boolean operators? Why did you do that?
This is part of the systematic review process. More information can be found here: https://environmentalevidence.org/information-for-authors/5-conducting-a-search/
- 305 – could you simply say “Geographic location was constrained to the region between the Isthmus of Tehuantepec and the Isthmus of Panama.”?
The authors wanted to make clear that we only included terrestrial palaeoenvironmental proxy evidence, hence the added detail.
- 306 – change to “10 August 2020”
The authors have updated the manuscript with the reviewer’s suggestions.
- 314 – Not sure what is meant by “Search strings were refined several times to remove terms containing generic nomenclature while still maintaining all relevant papers.”
This is part of the systematic review process. More information can be found here: https://environmentalevidence.org/information-for-authors/5-conducting-a-search/
- 318 – Also confused by “Search terms were iteratively modified and optimised until the entire Test Library was returned.”
This is part of the systematic review process. More information can be found here: https://environmentalevidence.org/information-for-authors/5-conducting-a-search/
- Table 4 – “Environmental proxy data must be representative of a point in time.” and “Environmental proxy data must be representative of different a point in time.” Maybe mention why these are criteria for inclusion.
This is part of the systematic review process. More information can be found here: https://environmentalevidence.org/information-for-authors/6-eligibility-screening/
- 327-336 – I was confused by the mix of past and future tense.
The authors have updated the manuscript with the reviewer’s suggested corrections.
- 355 – “cross cutting lens”?
The filter cross cutting lens is explained in table 5.
- Table 5 and elsewhere – keep in mind that “data are”
A matter of style – we will seek guidance from the Editors.
- 359 – “Metadata documenting the reproducibility of the study were extracted to give an indication of the quality of the research design; however, in common with most systematic evidence maps, critical appraisal of all studies selected at full-text was not undertaken and no studies were excluded from the evidence base on the basis of likelihood of bias.” I am unclear about how metadata document the reproducibility of a study.
Metadata, in systematic evidence evaluations, refer to any information about the included articles/studies – including information about included/excluded supplementary materials and raw data. Please see corresponding raw data included with the manuscript.
- 4 Note that the number of “Duplicates removed” in the figure (3508) is different from the value on line 366 of the text (3502).
The authors have corrected the typo in the manuscript.
- Section 3.3 – I would get rid of all the “#” and either use nothing or “n=XXX”
The authors have updated the manuscript with the reviewer’s suggested corrections.
- Figure 6 legend – “Number of studies extracted from different environmental archives.” I don’t think studies are extracted from archives.
“Studies” refer to individual proxy data sets – clarification has been made on L375.
- 429 – “The number of palaeoenvironmental records available increases throughout the Holocene.” I don’t think this is what you mean to say (clearly most of the studies only became available in the last few decades). So reword for clarity.
The authors have updated the manuscript and reworded the sentence for clarity.
- 434 – “These studies are considered low risk in terms of good methodological practice for sequence dating.” I am not familiar with this jargon, but what is meant by “risk” in this context?
Risk of time sequence error – avoiding extrapolation of a sequence outside of independent dates. The authors have clarified the text in the manuscript with reference to poorly, moderately and well constrained studies.
- 438 – “High risk studies (#205, 36.4%) are studies which have not dated the bottom of their sequence (Figure 9A). This is particularly problematic for studies which have been presented in their publications as measurements of depth only (#317; 58.9%) when they are discussed as time (Figure 9B), and this is further complicated when the raw study data is not available with the published article (#496, 92.2%) (Figure 9C).”
Risk of time sequence error – avoiding extrapolation of a sequence outside of independent dates. The authors have clarified the text in the manuscript with reference to poorly, moderately and well constrained studies.
- Change “which” to “that” in several places (lines 438-439). I am not sure “studies” date sequences. What does “This” refer to in the second sentence? Also “studies which have been presented in their publications as measurements of depth only” – do you mean “for studies in which published data are reported only with respect to depth in the sequence, but are discussed in terms of time?” Not really certain what was intended.
Please see text clarification in the manuscript.
- Figure 8. I think the legend need some clarification. Why is “proxies” in parentheses after “studies”? You should label the y axis. I gather that “forest, land use change, and water” are what was inferred in these studies.
“Proxies” has been deleted as the use of the word “studies” (for systematic evidence evaluations) has been clarified in L375.
- Figure 9. “Percentage of studies with a high, medium or low risk for sequence dating.” Again, I am unsure of what is meant by “risk.”
The authors have clarified the text in the manuscript.
- Table 6. I don’t think the header says what you want it to say. Maybe something like “Analyzed variable in paleoenvironmental archives and the environmental variable(s) for which it served as a proxy.” Also, I think there are only 17 such proxy variables, not 18, as Testate amoebae and Thecamoebians are one and the same.
The authors have updated the manuscript with the reviewer’s suggested corrections.
- 483 – Change “Relative” to “Relatively”
The authors have updated the manuscript with the reviewer’s suggested corrections.
- 488 – Change “while” to “whereas”
The authors have updated the manuscript with the reviewer’s suggested corrections.
- 499 – what is meant by “soil sediments”?
The authors report that proxies were extracted from soil samples (this is common in archaeological excavations).
- 510 – “There has been a particular increase in the number of studies published surrounding the last decade (49.5%).” What does that mean? Literally, it seems to me that these studies deal with changes between 2011 and today.
The authors have updated the manuscript to clarify the reviewer’s comment.
- 511 – Need a period before “Spatially”
The authors have updated the manuscript with the reviewer’s suggested corrections.
- 515 – “Evidence for forests are”?
Text changed to ‘Evidence for forests is…’
- 543 – “however, the absence of this data with the published article presents a worrying barrier to future access, use and quality assessment of these data sets.”?
The authors have updated the manuscript with the reviewer’s suggested corrections.
- 549 – “The guidelines and quality of dating practices for palaeoenvironmental research across the Central American Isthmus is recommended for review.” Not sure what you propose here. Are there “guidelines”?
The authors have clarified this sentence.
- 550 – “Accurately dating and presenting palaeoenvironmental studies, inclusive of the events represented within them, presents significant uncertainty.” So, if we ignore the middle clause, you are saying that “Accurately dating studies presents significant uncertainty.” How can that be? I would think that accurate dating would eliminate uncertainty. But furthermore, I do not think one “dates studies.”
The authors have clarified the text in the manuscript.
- 552 – “This is due to a number of factors including good methodological practice but also:” So I read that to mean that the “significant uncertainty” is caused by “good methodological practice.”
The authors have updated the manuscript with the reviewer’s suggested corrections.
- 556 – do you mean “variance” or “differences”?
The authors have updated the manuscript with the reviewer’s suggested corrections.
- 568 – “Identifying the impacts of different land uses across space and time would contribute towards our understanding of the long-term impacts to forest community composition and arboreal cover.” Yes, but somewhere I think it is critical to mention that the causes for shift in forest cover/composition today are very different from what they were in the past. Today the forest is removed with chain saws and much of the cleared land is used for cattle ranching. Pre-Columbian people worked with stone tools and fire, and there were no large domestic herbivores.
The authors agree with the reviewer but feel this is a discussion for a future paper.
- 585 – you do not need the hyphen with words ending in “ly”
This appears to be a formatting relic of the journal’s pdf creation. We will check carefully future pdfs generated in proofs.
- 606 – “to our understanding of different types of ecosystem resilience”? What are different types of ecosystem resilience?
The authors acknowledge that each of these sections could be vastly expanded (as they are intended to be the basis of future review papers, using this evidence base); however, the authors do not feel that expanding on these sections is necessary for this manuscript.
- 607 – “Using the palaeoecological records identified in this study, disturbance events and rates of recovery could be quantitatively identified and analysed from high resolution data sets using locally defined thresholds.” Not sure what you mean by “locally defined thresholds”? How will those be established?
Please see reference added.
- 614-619 – what does “key topics” refer to? “to inform on land-use policy”? I think you mean “to inform land-use policy”
The authors have clarified the text in the manuscript.
- “Integrating modern measurements, historical records, and palaeoenvironmnental data would enable quantifiable estimates of the above impacts and provide the knowledge base to make informed policy decisions in the future.” What does “above impacts” refer to? Aren’t “(a) climate change, (b) increasing populations pressures, and (c) land use change” the drivers of what you are interested in measuring in the paleo archives?
The authors have clarified the text in the manuscript.
- I hope that another reviewer, who is more familiar with the approaches used in this article, can weigh in. I think the piece could benefit from a complete overhaul, in which the objectives are clearly stated and the findings, re those objectives, are clearly presented. It is the prerogative of the authors to decide what their goals were in undertaking this study, but I think they muddy the waters by including boiler plate verbiage about how such paleo information can be used for risk mitigation, planning and conservation. Maybe I am just cynical, but I am dubious that the data can (or will be) used in such a fashion. The socio-political situation today in much of Central America will simply prevent the success of effective conservation efforts in many cases. I am a “believer” in Conservation Paleobiology, but its utility can be overestimated. In any case, is it not sufficient to say that you undertook this study to evaluate the information that is out there regarding forest dynamics in CA throughout the Holocene? Seems to me that the history of such tropical biotic changes is inherently interesting. And why don’t you get rid of “water” and “land use change” in the title? Aren’t those the drivers of changes in forest cover and composition? Maybe I am wrong, but it seems to me that the greatest rationale for this analysis of the existing literature was to identify what further work is required to build a clearer picture of Holocene vegetation change in Central America, and make suggestions about how that can be accomplished. Apparently, there are data gaps with respect to time intervals, geographic areas and forest types studied. But as I understand, one of the big problems is that raw data for most studies do not accompany the original publications. Do you suspect that access to that raw data would alleviate the need for more studies in neglected areas or time intervals? Anyway, I think considerable effort went into this analysis and I think it would be helpful if the findings were presented in a simple, straightforward way.
We thought the comments from a person from outside the systematic review community were very helpful in pointing out where clarification of the method is needed in the Special Issue overview, if not in the individual papers themselves. We followed the ROSES (and PRISMA) guidelines for reporting our findings and we feel these are relatively simple and straight forward to follow. We will be guided by the Special Issue Editors in adding as much additional information as is deemed helpful to the ‘lay reader’ (i.e. not familiar with SR methods) as we want our paper to have wide appeal and applicability.
Round 2
Reviewer 2 Report
This version is tremendously improved. It reads much more smoothly and is clearer. The only substantive issue I found is in the bar plot of Fig. 5B, in which Quaternary International appears twice. Also, note spelling of caribaea. I suggest some other minor edits in an attached pdf. There are some inconsistencies re use of accents (e.g. Panama), capitalization, and I am not sure about the guidelines for capitalization of section headers in the text. I leave that to the editors. Good job of re-working that. Mark Brenner

Author Response
Dear Professor Brenner,
The authors would like to thank you for your detailed review of our paper, and for providing both helpful and constructive comments. Your input has been greatly appreciated and we (the authors) would like to acknowledge your contribution to the improvement of our manuscript.
We have incorporated all of your requested changes.
Yours sincerely,
William J. Harvey